# Mixed Federated Learning: Joint Decentralized and Centralized Learning

## Abstract

Federated learning (FL) enables learning from decentralized privacy-sensitive data, with computations on raw data confined to take place at edge clients. This paper introduces *mixed FL*, which incorporates an additional loss term calculated at the coordinating server (while maintaining FL's private data restrictions). For example, additional datacenter data can be leveraged to jointly learn from centralized (datacenter) and decentralized (federated) training data and better match an expected inference data distribution. *Mixed FL* also enables offloading some intensive computations (e.g., embedding regularization) to the server, greatly reducing communication and client computation load. For these and other *mixed FL* use cases, we present three algorithms: Parallel Training, 1-way Gradient Transfer, and 2-way Gradient Transfer. We perform extensive experiments of the algorithms on three tasks, demonstrating that *mixed FL* can blend training data to achieve an oracle's accuracy on an inference distribution, and can reduce communication and computation overhead by more than 90%. Finally, we state convergence bounds for all algorithms, and give intuition on the *mixed FL* problems best suited to each. The theory confirms our empirical observations of how the algorithms perform under different *mixed FL* problem settings.

## 1 Introduction

Federated learning (FL) (McMahan et al., 2017) is a machine learning setting where multiple 'clients' (e.g., mobile phones) collaborate to train a model under coordination of a central server. Clients' raw data are never transferred. Instead, focused updates intended for immediate aggregation are used to achieve the learning objective (Kairouz et al., 2019). FL typically delivers model quality improvements because training examples gathered *in situ* by clients reflect actual inference serving requests. For example, a mobile keyboard next-word prediction model can be trained from actual SMS messages, yielding higher accuracy than a model trained on a proxy document corpus. Because of the benefits, FL has been used to train production models for many applications (Hard et al., 2018; Ramaswamy et al., 2019; Apple, 2019; Ramaswamy et al., 2020; Hartmann, 2021; Hard et al., 2022).

Building on FL, we can gain significant benefits from 'mixed FL': jointly[1] training with an additional *centralized* objective in conjunction with the *decentralized* objective of FL. Let $\boldsymbol{x}$ be model parameters to be optimized. Let $f$ denote a mixed loss, a sum[2] of a federated loss $f_{\text{f}}$ and a centralized loss $f_{\text{c}}$:

$$f(\boldsymbol{x}) = f_{\text{f}}(\boldsymbol{x}) + f_{\text{c}}(\boldsymbol{x}) \tag{1}$$

Mixed loss $f$ might be a more useful training objective than $f_{\text{f}}$ for many reasons, including:

**Mitigating Distribution Shift by Adding Centralized Data to FL**   While FL helps with reducing train vs. inference distribution skew, it may not remove it completely. Examples include: training device populations that are subsets of inference device populations (e.g., training on high-end phones, for eventual use also on low-end phones), label-biased example retention on edge clients (e.g., only retaining positive examples of a binary classification task), and infrequent safety-critical example

---

[1]We use 'joint' to distinguish our work from sequential 'central-then-FL' use cases, e.g. transfer learning.

[2]To simplify we subsume any relative weights into loss terms, i.e. this can be $f(\boldsymbol{x}) = (w_{\text{f}} \tilde{f}_{\text{f}}(\boldsymbol{x})) + (w_{\text{c}} \tilde{f}_{\text{c}}(\boldsymbol{x}))$.

events with outsized importance (e.g., automotive hard-braking events needed to train a self-driving AI) (Anonymous, a). The benefits of FL can be achieved while overcoming remaining distribution skew by incorporating data from an additional datacenter dataset, via mixed FL. This affords a composite set of training data that better matches the inference distribution.

**Reducing Client Computation and Communication**   In representation learning, negative examples are used to push dissimilar items apart in a latent embedding space while keeping positive examples closer together (Oord et al., 2018). In federated settings, clients' caches may have limited local negative examples, and recent work (Anonymous, b) showed this significantly degrades performance compared to centralized learning. This work also showed that using an additional loss (a regularization) to push representations apart, instead of negative examples, can resolve this performance gap. However, if done naively this requires communicating and computing over a large embedding table, introducing massive overhead for large-scale tasks. Applying mixed FL, where federated loss $f_\mathrm{f}$ is the primary 'affinity' loss and centralized loss $f_\mathrm{c}$ is the 'spreadout' regularization, avoids communicating the entire embedding table to clients and greatly reduces client computation.

Though mixed FL can clearly be useful, an actual process to minimize $f$ is not trivial. FL requires that clients' data stay on device, as they contain private information that possibly reveals personal identity. Moreover, centralized loss/data is expected to differ significantly[3] from client loss/data.

**Contributions**

- We motivate the mixed FL problem and present three algorithms for addressing it: PARALLEL TRAINING (PT), 1-WAY GRADIENT TRANSFER (1-W GT), and 2-WAY GRADIENT TRANSFER (2-W GT). These algorithms maintain the data privacy protections inherent in FL. [Section 2]
- We experiment with facial attribute classification and language modeling, demonstrating that our algorithms overcome distribution shift. We match the accuracy of hypothetical 'oracle' scenarios where the entire inference distribution was colocated for training. [Section 4]
- We experiment with user-embedding based movie recommendation, reducing communication overhead by 93.9% and client computation by 99.9% with no degradation in quality. [Section 4]
- We state convergence bounds for each algorithm (in strongly, general, and non-convex settings), providing theoretical explanations for convergence behaviors we observe in the experiments. This indicates how the algorithms will perform on new mixed FL tasks. [Section 5]

## 2   ALGORITHMS

In FL, the loss function $f_\mathrm{f}$ is an average of client loss functions $f_i$. The client loss $f_i$ is an expectation over batches of data examples $\mathcal{B}_i$ on client $i$.

$$f_\mathrm{f}(\boldsymbol{x}) = \frac{1}{N} \sum_{i=1}^{N} f_i(\boldsymbol{x}), \quad f_i(\boldsymbol{x}) = \mathbb{E}_{\mathcal{B}_i} \left[ f_i(\boldsymbol{x}; \mathcal{B}_i) \right] \tag{2}$$

FEDAVG (McMahan et al., 2017) is a ubiquitous, heuristic FL method designed to minimize Equation 2 w.r.t. model $\boldsymbol{x}$ in a manner that allows all client data ($\mathcal{B}_i$) to remain at respective clients $i$. Providing strong privacy protection is a major motivation for FL. Storing raw data locally on clients rather than replicating it on servers decreases the attack surface of the system. Also, using focused ephemeral updates and early aggregation follows principles of data minimization (White House Report, 2013).[4]

While training with loss $f_\mathrm{f}$ via FEDAVG can yield an effective model $\boldsymbol{x}$, this paper shows there are scenarios where 'mixing' in an additional 'centralized' loss $f_\mathrm{c}$ proves beneficial to the training of $\boldsymbol{x}$. Such a loss term can make use of batches of centralized data examples $\mathcal{B}_\mathrm{c}$, from a datacenter dataset:

$$f_\mathrm{c}(\boldsymbol{x}) = \mathbb{E}_{\mathcal{B}_\mathrm{c}} \left[ f_\mathrm{c}(\boldsymbol{x}; \mathcal{B}_\mathrm{c}) \right] \tag{3}$$

---

[3]Were they not to differ, one could treat a centralized compute node as an additional client in standard FL, and simply make use of an established FL algorithm like FEDAVG for training $\boldsymbol{x}$.

[4]Even stronger privacy properties are possible when FL is combined with technologies such as differential privacy (DP) and secure multiparty computation (SMPC) (Wang et al., 2021).

The centralized loss $f_{\mathrm{c}}$ will differ from the federated loss $f_{\mathrm{f}}$ (else it would not be useful). This can be a difference in the distributions that $\mathcal{B}_{\mathrm{c}}$ and $\mathcal{B}_i$ are drawn from, and/or in the functional forms of $f_{\mathrm{c}}$ and $f_i$. We will present an expression that quantifies the difference between $f_{\mathrm{c}}$ and $f_{\mathrm{f}}$ in Section 5.

We now state our mixed FL algorithms (Algorithms 1 and 2). Appendix A has a few practical details.

- **PARALLEL TRAINING** performs a round of FEDAVG (minimizing $f_{\mathrm{f}}$) in parallel with steps of centralized training (minimizing $f_{\mathrm{c}}$), merges (e.g., averages) the respective updates, and repeats. Green in Algorithm 1 indicates added steps beyond FEDAVG for PARALLEL TRAINING.
- **1-WAY GRADIENT TRANSFER** starts a round by calculating a gradient of $f_{\mathrm{c}}$. It is sent to participating clients and summed with clients' gradients of $f_i$ during client optimization. Blue in Algorithm 2 indicates added steps beyond FEDAVG for 1-WAY GRADIENT TRANSFER.
- **2-WAY GRADIENT TRANSFER** is PARALLEL TRAINING with gradient sharing. Two gradients are now used, one based on $f_{\mathrm{c}}$ and sent to clients (like 1-W GT), one based on $f_{\mathrm{f}}$ and applied centrally. Purple in Algorithm 1 is added steps beyond PT for 2-WAY GRADIENT TRANSFER.

---

**Algorithm 1:** PARALLEL TRAINING and 2-WAY GRADIENT TRANSFER
(FEDAVG with added steps for PARALLEL TRAINING and further steps for 2-WAY GRADIENT TRANSFER)

**Input:** Initial model $\boldsymbol{x}^{(0)}$; CLIENTOPTIMIZER, SERVEROPTIMIZER, CENTRALOPTIMIZER, MERGEOPTIMIZER with respective
  learning rates $\eta, \eta_{\mathrm{s}}, \eta_{\mathrm{c}}, \eta_{\mathrm{m}}$; central loss function $f_{\mathrm{c}}$ (3); initial augmenting centralized/federated gradients, $\tilde{g}_{\mathrm{c}}^{(0)}, \tilde{g}_{\mathrm{f}}^{(0)}$ (zeroed)
**for** $t \in \{0, 1, \ldots, T-1\}$ **do**
  Initialize central model $\boldsymbol{x}_{\mathrm{c}}^{(t,0)} = \boldsymbol{x}^{(t)}$
  **for central step** $k = 0, \ldots, K_c - 1$ **do**
    Sample centralized batch $\mathcal{B}_{\mathrm{c}}^{(k)}$; compute stochastic gradient $g_{\mathrm{c}}(\boldsymbol{x}_{\mathrm{c}}^{(t,k)}; \mathcal{B}_{\mathrm{c}}^{(k)})$ of $f_{\mathrm{c}}(\boldsymbol{x}_{\mathrm{c}}^{(t,k)})$
    Perform central update $\boldsymbol{x}_{\mathrm{c}}^{(t,k+1)} = \text{CENTRALOPTIMIZER}(\boldsymbol{x}_{\mathrm{c}}^{(t,k)}, g_{\mathrm{c}}(\boldsymbol{x}_{\mathrm{c}}^{(t,k)}; \mathcal{B}_{\mathrm{c}}^{(k)}) + \tilde{g}_{\mathrm{f}}^{(t)}, \eta_{\mathrm{c}}, t)$
  Compute central model delta $\boldsymbol{\Delta}_{\mathrm{c}}^{(t)} = \boldsymbol{x}_{\mathrm{c}}^{(t,K_c)} - \boldsymbol{x}^{(t)}$
  Sample a subset $\mathcal{S}^{(t)}$ of clients; **for client** $i \in \mathcal{S}^{(t)}$ **in parallel do**
    $\boldsymbol{\Delta}_i^{(t)}, p_i = \text{CLIENTUPDATE}(\boldsymbol{x}^{(t)}, \tilde{g}_{\mathrm{c}}^{(t)}, \text{CLIENTOPTIMIZER}, \eta, t)$
  Aggregate client changes $\boldsymbol{\Delta}^{(t)} = \sum_{i \in \mathcal{S}^{(t)}} p_i \boldsymbol{\Delta}_i^{(t)} / \sum_{i \in \mathcal{S}^{(t)}} p_i$
  Compute federated model $\boldsymbol{x}_{\mathrm{f}}^{(t)} = \text{SERVEROPTIMIZER}(\boldsymbol{x}^{(t)}, -\boldsymbol{\Delta}^{(t)}, \eta_{\mathrm{s}}, t)$
  Compute federated model delta $\boldsymbol{\Delta}_{\mathrm{f}}^{(t)} = \boldsymbol{x}_{\mathrm{f}}^{(t)} - \boldsymbol{x}^{(t)}$
  Aggregate central model and federated model deltas $\boldsymbol{\Delta}^{(t)} = \boldsymbol{\Delta}_{\mathrm{c}}^{(t)} + \boldsymbol{\Delta}_{\mathrm{f}}^{(t)}$
  Update global model $\boldsymbol{x}^{(t+1)} = \text{MERGEOPTIMIZER}(\boldsymbol{x}^{(t)}, -\boldsymbol{\Delta}^{(t)}, \eta_{\mathrm{m}}, t)$
  Update augmenting centralized gradient $\tilde{g}_{\mathrm{c}}^{(t+1)} = -\boldsymbol{\Delta}_{\mathrm{c}}^{(t)} / (\eta_{\mathrm{c}} K_c) - \tilde{g}_{\mathrm{f}}^{(t)}$
  Update augmenting federated gradient $\tilde{g}_{\mathrm{f}}^{(t+1)} = -\sum_{i \in \mathcal{S}^{(t)}} \boldsymbol{\Delta}_i^{(t)} / (\eta \sum_{i \in \mathcal{S}^{(t)}} K_i) - \tilde{g}_{\mathrm{c}}^{(t)}$   (*see Appendix A*)

CLIENTUPDATE:
**Input:** Initial client model $\boldsymbol{x}_i^{(t,0)}$; (possible) augmenting gradient $\tilde{g}_{\mathrm{c}}^{(t)}$; CLIENTOPTIMIZER, learning rate $\eta$; round $t$; client loss $f_i$ (2)
Initialize client weight $p_i = 0$
**for client step** $k = 0, \ldots, K_i - 1$ **do**
  Sample batch $\mathcal{B}_i^{(k)}$; compute stochastic gradient $g_i(\boldsymbol{x}_i^{(t,k)}; \mathcal{B}_i^{(k)})$ of $f_i(\boldsymbol{x}_i^{(t,k)})$; update client weight $p_i = p_i + |\mathcal{B}_i^{(k)}|$
  Perform client update $\boldsymbol{x}_i^{(t,k+1)} = \text{CLIENTOPTIMIZER}(\boldsymbol{x}_i^{(t,k)}, g_i(\boldsymbol{x}_i^{(t,k)}; \mathcal{B}_i^{(k)}) + \tilde{g}_{\mathrm{c}}^{(t)}, \eta, t)$
Compute client model changes $\boldsymbol{\Delta}_i^{(t)} = \boldsymbol{x}_i^{(t,K_i)} - \boldsymbol{x}_i^{(t,0)}$ and **return** $\boldsymbol{\Delta}_i^{(t)}, p_i$

---

**Algorithm 2:** 1-WAY GRADIENT TRANSFER (FEDAVG (McMahan et al., 2017) with added steps)

**Input:** Initial model $\boldsymbol{x}^{(0)}$; CLIENTOPTIMIZER, SERVEROPTIMIZER with respective learning rates $\eta, \eta_{\mathrm{s}}$; central loss function $f_{\mathrm{c}}$ (3)
**for** $t \in \{0, 1, \ldots, T-1\}$ **do**
  Sample batch $\mathcal{B}_{\mathrm{c}}^{(t)}$; compute stochastic gradient $g_{\mathrm{c}}(\boldsymbol{x}^{(t)}; \mathcal{B}_{\mathrm{c}}^{(t)})$ of $f_{\mathrm{c}}(\boldsymbol{x}^{(t)})$; set augmenting gradient $\tilde{g}_{\mathrm{c}}^{(t)} = g_{\mathrm{c}}(\boldsymbol{x}^{(t)}; \mathcal{B}_{\mathrm{c}}^{(t)})$
  Sample a subset $\mathcal{S}^{(t)}$ of clients; **for client** $i \in \mathcal{S}^{(t)}$ **in parallel do**
    $\boldsymbol{\Delta}_i^{(t)}, p_i = \text{CLIENTUPDATE}(\boldsymbol{x}^{(t)}, \tilde{g}_{\mathrm{c}}^{(t)}, \text{CLIENTOPTIMIZER}, \eta)$   (CLIENTUPDATE *function defined in Algorithm 1*)
  Aggregate client changes $\boldsymbol{\Delta}^{(t)} = \sum_{i \in \mathcal{S}^{(t)}} p_i \boldsymbol{\Delta}_i^{(t)} / \sum_{i \in \mathcal{S}^{(t)}} p_i$
  Update global model $\boldsymbol{x}^{(t+1)} = \text{SERVEROPTIMIZER}(\boldsymbol{x}^{(t)}, -\boldsymbol{\Delta}^{(t)}, \eta_{\mathrm{s}}, t)$

---

## 3  RELATED WORK

There are parallels between GRADIENT TRANSFER and algorithms addressing inter-client data heterogeneity in FL, like SCAFFOLD (Karimireddy et al., 2020b) or Mime (Karimireddy et al., 2020a). Those algorithms calculate a gradient reflective of the entire federated population and transmit it to clients to reduce update variance, improving optimization on non-IID client data. GRADIENT TRANSFER calculates a gradient reflective of *centralized* data/loss, to augment client computations

of *decentralized* data/loss (in 2-w GT, also the converse). However, SCAFFOLD requires keeping state at the server (control variates) for *each* participating client, impractical in real large-scale FL systems (Kairouz et al., 2019). 2-w GT only requires state (augmenting gradients) for two entities, and so is easily implemented.

Another instance of a non-IID client approach (Zhao et al., 2018) influencing mixed FL is the EXAMPLE TRANSFER algorithm (Anonymous, a). Here, centralized examples are sent to federated clients, instead of gradients. This is typically precluded in real FL applications. The data volume needed to transfer may be excessive, and in general EXAMPLE TRANSFER does not offer solutions for one of the main motivations of this work, reducing client computation and communication.

'Transfer learning' also involves two different training datasets, but has a different purpose. Transfer learning pretrains a model on a distribution (e.g., centralized data in a datacenter), then fine-tunes it on the actual distribution of interest (e.g., decentralized data via FL). It is desirable as a way to quickly train on the latter distribution (e.g., as in Ro et al. (2022)). But its sequential approach results in *catastrophic forgetting* (McCloskey & Cohen, 1989; Ratcliff, 1990; French, 1999); accuracy on pretraining data is lost as the model learns to fit fine-tuning data instead. In mixed FL, we desire good inference performance on *all* data distributions trained on. See Appendix B.4.5 for more.

In differentially private (DP) optimization, a line of work has aimed to improve privacy/utility tradeoffs by utilizing additional non-private data. One way is to use non-private data to pretrain (Abadi et al., 2016). Another avenue is to use non-private data to learn the gradient geometry (Zhou et al., 2020; Amid et al., 2021; Asi et al., 2021; Kairouz et al., 2021a; Li et al., 2022), improving accuracy by enabling tighter, non-isotropic gradient noise during DP optimization. Amid et al. (2021) and Li et al. (2022) consider the FL use case[5]. As in transfer learning, additional data is used only to improve performance on a single distribution, and retaining accuracy on other distributions is a non-goal (in contrast to mixed FL). Also, the non-private data used is generally matching (in distribution) to the private data, whereas in mixed FL we typically explicitly leverage *distinct* distributions.

A few recent works (Yu et al., 2020; Elbir et al., 2021; Jeong et al., 2021) present instances of mixed FL problems; they shift computations to the server that are intensive or impossible at the clients. These works do not address distribution shift or present more general mixed FL algorithms.

## 4 EXPERIMENTS

We now present experiments on three tasks (Table 1), showing a range of problems for which mixed FL is useful. They also show the comparative performance of each algorithm described in Section 2.

Table 1: Experiments summary. For training hyperparameters and further details, see Appendix B.

|  | SMILE CLASSIFICATION | LANGUAGE MODELING | MOVIE RECOMMEND. |
|---|---|---|---|
| MODEL ARCH. TYPE | FULLY-CONNECTED | RNN | DUAL ENCODER |
| FEDERATED DATA | CELEBA (SMILING) | STACK OVERFLOW | MOVIELENS |
| FED. CLIENT LOSS ($f_i$) | BINARY C.E. | CATEGORICAL C.E. | HINGE |
| FED. WEIGHT ($w_\mathrm{f}$) | 0.5 | 0.75 | 0.5 |
| CENTRALIZED DATA | CELEBA (NON-SMILING) | WIKIPEDIA | - |
| CENT. LOSS ($f_\mathrm{c}$) | BINARY C.E. | CATEGORICAL C.E. | SPREADOUT (REG.) |
| CENT. WEIGHT ($w_\mathrm{c}$) | 0.5 | 0.25 | 0.5 |

### 4.1 ADDRESS LABEL IMBALANCE IN TRAIN DATA (SMILE CLASSIFIER; CELEBA)

Earlier work (Anonymous, a) motivated mixed FL with the example problem of training a 'smiling'-vs.-'unsmiling' classifier via FL with mobile phones, with the challenge that the phones' camera application (by the nature of its usage) tends to only persist images of smiling faces. A solution

---

[5]An interesting similarity between PDA-DPMD (Amid et al., 2021) and our work: in PDA-DPMD for FL, a first order approximation of mirror descent is used, where the server model update is calculated as weighted sum of private (federated) and public loss terms, just as in PARALLEL TRAINING or 2-WAY GRADIENT TRANSFER.

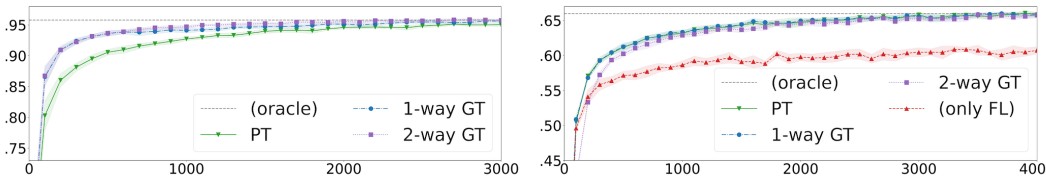

(a) Smile Classifier: Eval. AUC (ROC) vs. Round.    (b) Language Model: Evaluation Accuracy vs. Round.

Figure 1: Mixed FL resolves distribution shift, enabling accuracy equal to if data were colocated and centrally trained ('oracle'). The smile classifier reaches an oracle's evaluation AUC of ROC of over 0.95. The language model reaches an oracle's evaluation accuracy of over 0.66. Evaluation is over *all* data (i.e., smiling *and* non-smiling faces; Stack Overflow *and* Wikipedia). Plots show 95% conf.

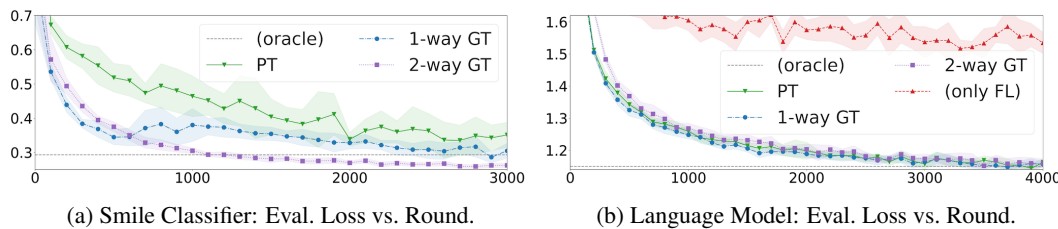

(a) Smile Classifier: Eval. Loss vs. Round.    (b) Language Model: Eval. Loss vs. Round.

Figure 2: Comparative convergence differs; (a) shows PT converging worse than 1-w GT or 2-w GT, while (b) shows all algorithms converging the same. See Section 5.3 for theoretical explanation.

for this severe label imbalance is to apply mixed FL, utilizing an additional datacenter dataset of unsmiling faces to train a capable classifier. To experiment, CelebA data[6] (Liu et al., 2015; Caldas et al., 2018) is split into a federated 'smiling' dataset and centralized 'unsmiling' dataset.

Figures 1a and 2a show the AUC and loss convergence of our three algorithms when applied to this problem. Empirically, we observe GRADIENT TRANSFER converge faster than PARALLEL TRAINING. Section 5 will provide a theoretical explanation for this behavior.

## 4.2 MITIGATE BIAS IN TRAIN DATA (LANGUAGE MODEL; STACK OVERFLOW, WIKIPEDIA)

Consider the problem of learning a language model like a RNN-based next character prediction model, used to make typing suggestions to a user in a mobile keyboard application. Because the ultimate inference application is on mobile phones, it is natural to train this model via FL, leveraging cached SMS text content highly reflective of inference time usage (at least for some users).

However, the mobile phones participating in the federated learning of the model might be only a subset of the mobile phones for which we desire to deploy for inference. Higher-end mobile phones can disproportionately participate in FL, as their larger memory and faster processors allow them to complete client training faster. But to do well at inference, a model should make accurate predictions for users of lower-end phones as well. A purely FL approach can do an inadequate job of learning these users' usage patterns. (See Kairouz et al. (2019) for more on aspects of fairness and bias in FL.)

Mixed FL overcomes this problem, by training a model jointly on federated data (representative of users of higher-end phones) and a datacenter dataset (representative of users of lower-end phones). We simulate this scenario using two large public datasets: the Stack Overflow dataset[7] (Kaggle) for federated data, and the Wikipedia dataset[8] (Wikimedia Foundation) for datacenter data. Figure 1b shows results. The 'only FL' scenario learns Stack Overflow (but *not* Wikipedia) patterns of character usage, and so has reduced accuracy ($\sim 0.60$) when evaluated on examples from both datasets. The

---

[6]CelebA federated data available via open source FL software (TFF CelebA documentation, 2022).

[7]Stack Overflow federated data available via open source (TFF StackOverflow documentation, 2022).

[8]Wikipedia data available via open source (TFDS Wikipedia (20201201.en) documentation, 2022).

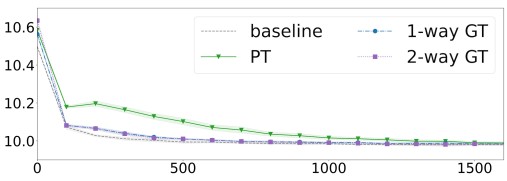

Figure 3: Movie prediction, Eval. Loss vs. Round. We see that mixed FL results in similar loss as the more expensive baseline scenario (see Table 2).

Table 2: Movie recommendation: computation (COMP.) and communication (COMM.) overhead per client. The baseline scenario computes everything on clients. See Appendix B.3 for analysis.

|  | COMP. (MFLOP) | COMM. (KB) |
|---|---|---|
| BASELINE | 125.16 | 494 |
| PT | 0.025 | 20 |
| 1-W GT | 0.025 | 30 |
| 2-W GT | 0.025 | 30 |

mixed FL algorithms demonstrate learning both: they all achieve an evaluation accuracy ($\sim 0.66$) comparable to an imagined 'oracle' that could centrally train on the combination of datasets.

Unlike the smile classification experiment, here PARALLEL TRAINING converges about as fast as GRADIENT TRANSFER. Section 5 will provide a theoretical explanation for this behavior.

### 4.3 REGULARIZE EMBEDDINGS AT SERVER (MOVIE RECOMMENDATION; MOVIELENS)

The third task we study, movie recommendation, is one with an embedding regularization term as described in Section 1. As Table 1 shows, a key difference from the previous two mixed FL experimental scenarios is that here we are mixing losses with different functional forms (instead of mixing different datacenter and federated data distributions). We study this scenario by training a dual encoder representation learning model (Covington et al., 2016) for next movie prediction on the MovieLens dataset (Harper & Konstan, 2015; GroupLens).

As described in Section 1, limited negative examples can degrade representation learning performance. Previous work (Anonymous, b) proposed using losses insensitive to local client negatives to improve federated model performance. They observed significantly improved performance by using a two-part loss: (1) a hinge loss to pull embeddings for similar movies together, and (2) a spreadout regularization (Zhang et al., 2017) to push embeddings for unrelated movies apart. For clients to calculate (2), the server must communicate all movie embeddings to each client, and clients must perform a matrix multiplication over the entire embedding table. This introduces enormous computation and communication overhead when the number of movies is large.

Mixed FL can alleviate this communication and computation overhead. Instead of computing both loss terms on clients, clients calculate only the hinge loss and the server calculates the expensive regularization, avoiding costly computation on each client. Also, computing the hinge loss only requires the embeddings of movies in a client's local dataset. Federated select (Charles et al., 2022) enables *only* those embeddings to be sent to that client, saving communication and on-client memory.

Experiments show all mixed FL algorithms achieve model performance (around 0.1 for recall@10) comparable to the baseline scenario where everything is computed on the clients. Moreover, mixed FL eliminates more than 99.9% of client computation and more than 93.9% of communication (see Table 2). For computation and communication analysis, see Appendix B. Note that real-world models can be much larger than this movie recommendation model[9]. Without mixed FL, communicating such large models to clients and computing regularization would be impractical in large-scale settings. PARALLEL TRAINING converges slightly slower than either GRADIENT TRANSFER algorithm (Figure 3) but reaches the same evaluation loss at around 1500 rounds.

## 5 CONVERGENCE

### 5.1 PRELIMINARIES

We now describe convergence properties for each mixed FL algorithm from Section 2. We assume mixed loss $f$ has a finite minimizer (i.e., $\exists \, \boldsymbol{x}^* \, s.t. \, f(\boldsymbol{x}) \geq f(\boldsymbol{x}^*) \, \forall \, \boldsymbol{x}$). We assume client losses $f_i$

---

[9]E.g., for a next URL prediction task with millions of URLs the embedding table size can reach gigabytes.

and centralized loss $f_c$ are $\beta$-smooth. If $f_i$ are $\beta$-smooth, federated loss $f_f$ is also[10]. For some results, we assume $f_i$ and $f_c$ are $\mu$-convex (possibly strongly convex, $\mu > 0$). If $f_i$ are $\mu$-convex, $f_f$ is also[11].

For a parameter vector $\boldsymbol{x}$, we use $\nabla f_i(\boldsymbol{x})$ to denote the full gradient of $f_i$ (i.e., over all data on client $i$). Similarly, $\nabla f_f(\boldsymbol{x})$ and $\nabla f_c(\boldsymbol{x})$ denote full gradients[12] of $f_f$ and $f_c$ at $\boldsymbol{x}$. We use $g_i(\boldsymbol{x})$ to denote an unbiased *stochastic* gradient of $f_i$, calculated on a random batch $\mathcal{B}_i$ of examples on client $i$.

We focus on the impact to convergence when differences exist between the federated and centralized losses/data. As such, we make the following homogeneity assumption about the federated data, which simplifies the analysis and brings out the key differences. Our analysis can be easily extended to heterogeneous clients by assuming a bound on variance of the client gradients.

**Assumption 5.1.** The federated clients have homogeneous data distributions (i.e., with examples that are drawn IID from a common data distribution), and their stochastic gradients have bounded variance. Specifically, for some $\sigma > 0$, we have for all clients $i$ and parameter vectors $\boldsymbol{x}$:

$$\mathbb{E}\left[g_i(\boldsymbol{x})\right] = \nabla f_f(\boldsymbol{x}), \quad \mathbb{E}\left\|g_i(\boldsymbol{x}) - \nabla f_f(\boldsymbol{x})\right\|^2 \leq \sigma^2. \tag{4}$$

Under such conditions, FEDAVG convergence can match SGD (Table 2, Karimireddy et al. (2020b)).

Let $g_f$ denote an unbiased stochastic gradient of federated loss $f_f$, formed by randomly sampling a cohort of $S$ (out of $N$ total) clients, randomly sampling a batch $\mathcal{B}_i$ of client data examples, and averaging the respective client stochastic gradients. Given Assumption 5.1 we bound variance of $g_f$:

$$g_f(\boldsymbol{x}) = \frac{1}{S}\sum_{i \in \mathcal{S}} g_i(\boldsymbol{x}), \quad \mathbb{E}\left\|g_f(\boldsymbol{x}) - \nabla f_f(\boldsymbol{x})\right\|^2 = \mathbb{E}\left\|\frac{1}{S}\sum_{i \in \mathcal{S}} g_i(\boldsymbol{x}) - \nabla f_f(\boldsymbol{x})\right\|^2 \leq \frac{1}{S}\sigma^2 \tag{5}$$

Let $g_c(\boldsymbol{x})$ denote a stochastic gradient of the centralized loss $f_c$ at $\boldsymbol{x}$, calculated on a randomly sampled batch $\mathcal{B}_c$ of centralized examples (from a datacenter dataset), with variance bounded by $\sigma_c^2$:

$$\mathbb{E}\left\|g_c(\boldsymbol{x}) - \nabla f_c(\boldsymbol{x})\right\|^2 \leq \sigma_c^2. \tag{6}$$

Summarizing Equations 4-6, a client's stochastic gradient $g_i(\boldsymbol{x})$ has variance bounded by $\sigma^2$, the federated cohort's stochastic gradient $g_f(\boldsymbol{x})$ has variance bounded by $\sigma^2/s$, and the centralized stochastic gradient $g_c(\boldsymbol{x})$ has variance bounded by $\sigma_c^2$. Increasing client batch size $|\mathcal{B}_i|$ reduces variance of $g_i(\boldsymbol{x})$ and $g_f(\boldsymbol{x})$, increasing cohort size $S$ reduces variance of $g_f(\boldsymbol{x})$, and increasing central batch size $|\mathcal{B}_c|$ reduces variance of $g_c(\boldsymbol{x})$.

Note that $\sigma^2/s$ only bounds variance *within* the federated data distribution, and $\sigma_c^2$ only bounds variance *within* the central data distribution. To say something about variance *across* the two data distributions, we adapt the notion of 'bounded gradient dissimilarity' (or 'BGD') introduced in Karimireddy et al. (2020b) (Definition A1), and apply it to the mixed FL scenario here.

**Definition 5.2** (mixed FL $(G, B)$-BGD)**.** There exist constants $G \geq 0$ and $B \geq 1$ such that $\forall \boldsymbol{x}$:

$$w_f\left\|\frac{\nabla f_f(\boldsymbol{x})}{w_f}\right\|^2 + w_c\left\|\frac{\nabla f_c(\boldsymbol{x})}{w_c}\right\|^2 \leq G^2 + B^2\left\|\nabla f(\boldsymbol{x})\right\|^2$$

In the definition, $w_f$ and $w_c$ are proportions of influence ($w_f + w_c = 1$) of the federated and centralized objectives on the overall mixed optimization. (The simplest setting is $w_f = w_c = 1/2$.)

## 5.2 BOUNDS

We can now state upper bounds on convergence (to an error in mixed loss smaller than $\epsilon$) for the respective mixed FL algorithms. For ease of comparison, the convergence bounds are summarized in Table 3. The Theorems and Proofs of these convergence bounds are given in Appendix C. As mentioned

---

[10]By its definition in Equation 2 combined with the triangle inequality.

[11]By its definition in Equation 2, it is convex combination of $f_i$.

[12]Note: $\nabla f_f(\boldsymbol{x})$ is useful for theoretical convergence analysis, but cannot be practically computed in a real cross-device FL setting. In contrast, $\nabla f_c(\boldsymbol{x})$ *can* be computed.

Table 3: Order of number of rounds required to be within $\epsilon$ of optimal mixed loss, for different mixing strategies. See Appendix C. $\sigma^2$ as in (4), $\sigma_c^2$ as in (6), $G$ and $B$ as in Def. 5.2 with $w_f = w_c = 1/2$. $\beta$ is smoothness (Def. D.1), $\mu$ is convexity bound (Def. D.4). $K$ is client local steps taken ($\geq 2$), $S$ is client cohort size (per round). $D$ and $F$ are initial distances/loss errors, described in Appendix C.

| | PARALLEL TRAINING | 1-W GT | 2-W GT |
|---|---|---|---|
| $\mu$-CONVEX | $\frac{(\sigma^2+S\sigma_c^2)}{KS\mu\epsilon} + \frac{G\sqrt{\beta}}{\mu\sqrt{\epsilon}} + \frac{B^2\beta}{\mu}\log(\frac{1}{\epsilon})$ | $\frac{(\sigma^2+KS\sigma_c^2)}{KS\mu\epsilon} + \frac{\beta}{\mu}\log(\frac{1}{\epsilon})$ | $\frac{(\sigma^2+S\sigma_c^2)}{KS\mu\epsilon} + \frac{\beta}{\mu}\log(\frac{1}{\epsilon})$ |
| CONVEX | $\frac{(\sigma^2+S\sigma_c^2)D^2}{KS\epsilon^2} + \frac{G\sqrt{\beta}}{\epsilon^{\frac{3}{2}}} + \frac{B^2\beta D^2}{\epsilon}$ | $\frac{(\sigma^2+KS\sigma_c^2)D^2}{KS\epsilon^2} + \frac{\beta D^2}{\epsilon}$ | $\frac{(\sigma^2+S\sigma_c^2)D^2}{KS\epsilon^2} + \frac{\beta D^2}{\epsilon}$ |
| NONCONVEX | $\frac{(\sigma^2+S\sigma_c^2)\beta F}{KS\epsilon^2} + \frac{G\sqrt{\beta}}{\epsilon^{\frac{3}{2}}} + \frac{B^2\beta F}{\epsilon}$ | $\frac{(\sigma^2+KS\sigma_c^2)\beta F}{KS\epsilon^2} + \frac{\beta F}{\epsilon}$ | $\frac{(\sigma^2+S\sigma_c^2)\beta F}{KS\epsilon^2} + \frac{\beta F}{\epsilon}$ |

previously, the analysis extends in a straightforward manner to the setting of heterogeneous clients assuming a bound on the variance of client gradients: for all $x$, $\frac{1}{N}\sum_{i=1}^{N}\|\nabla f_i(x) - \nabla f_f(x)\|^2 \leq \sigma_f^2$ for some $\sigma_f \geq 0$. Under this assumption, the bounds in Table 3 change by an additional $K\sigma_f^2$ term in the expression involving $\sigma^2$ and $\sigma_c^2$ in the parenthesis on the numerator of the leading term. The derivation of these more general bounds follows on the same lines, so we omit it for brevity.

Analyzing Table 3, there are several implications to be drawn.

**Significant ($G$,$B$)-BGD impedes PARALLEL TRAINING** The convergence bounds for PARALLEL TRAINING show a dependence on the $G$ and $B$ parameters from Definition 5.2. If a mixed FL problem involves a large amount of dissimilarity between the federated and centralized gradients (i.e., if $G \gg 0$ or $B \gg 1$), then PARALLEL TRAINING will be slower to converge than alternatives.

**Significant $\sigma_c^2$ impedes 1-WAY GRADIENT TRANSFER** 1-WAY GRADIENT TRANSFER is more sensitive to central variance $\sigma_c^2$. Unlike the other algorithms, the impact of $\sigma_c^2$ on convergence scales with the number of steps $K$. 1-WAY GRADIENT TRANSFER requires a central batch size $|\mathcal{B}_c|$ that is $K$ times larger to achieve the same impact on convergence. Intuitively, this makes sense; in a round, PARALLEL TRAINING and 2-WAY GRADIENT TRANSFER sample $K$ fresh batches during centralized optimization, while 1-WAY GRADIENT TRANSFER only samples a single central batch.

**2-WAY GRADIENT TRANSFER should always converge at least as well as others** The convergence bound for 2-WAY GRADIENT TRANSFER is unaffected by gradient dissimilarity (i.e., $G \gg 0$ or $B \gg 1$), unlike PARALLEL TRAINING. Also, the bound for 2-WAY GRADIENT TRANSFER is less sensitive to $\sigma_c^2$ than the bound for 1-WAY GRADIENT TRANSFER (as described above).

### 5.3 ANALYSIS OF EXPERIMENTS

PARALLEL TRAINING has a convergence bound substantially different than the GRADIENT TRANSFER algorithms; the dependence on the BGD parameters $G$ and $B$ indicates there are mixed FL problems where PARALLEL TRAINING is slower to converge than GRADIENT TRANSFER (in either form). How can we know if a particular problem is one where PARALLEL TRAINING will have slower convergence? It would be useful to know $G$ and $B$, but they cannot be exactly measured. $G$ and $B$ (Definition 5.2) are upper bounds holding $\forall x$, and the entire space of $x$ cannot realistically be checked. Instead, we introduce sampled approximations to empirically estimate these upper bounds.

Let $x^{(t)}$ be the global model at start of round $t$. Let $\tilde{\nabla}f_{f_t}, \tilde{\nabla}f_{c_t}, \tilde{\nabla}f_t$ be approximations of federated, centralized, total gradients at round $t$. Considering Definition 5.2, we define $\tilde{G}_t$ as a sampled approximation of $G$ assuming $B = 1$, and $\tilde{B}_t$ as a sampled approximation of $B$ assuming $G = 0$:

$$\tilde{\nabla}f_{f_t} = \frac{1}{S}\sum_{i\in\mathcal{S}}\left(g_i(x^{(t)})\right), \quad \tilde{\nabla}f_{c_t} = g_c(x^{(t)}), \quad \tilde{\nabla}f_t = \tilde{\nabla}f_{f_t} + \tilde{\nabla}f_{c_t}$$

$$\tilde{G}_t^2 = \frac{1}{w_f}\left\|\tilde{\nabla}f_{f_t}\right\|^2 + \frac{1}{w_c}\left\|\tilde{\nabla}f_{c_t}\right\|^2 - \left\|\tilde{\nabla}f_t\right\|^2, \quad \tilde{B}_t^2 = \left(\frac{1}{w_f}\left\|\tilde{\nabla}f_{f_t}\right\|^2 + \frac{1}{w_c}\left\|\tilde{\nabla}f_{c_t}\right\|^2\right) / \left\|\tilde{\nabla}f_t\right\|^2$$

$$(7)$$

Table 4: Sampled approximations of $(G,B)$-BGD. (See Appendix B.1 for details.)

|  | SMILE CLASSIFICATION | LANGUAGE MODELING | MOVIE RECOMMEND. |
|---|---|---|---|
| $\max_{10 \leq t \leq 100} \tilde{G}_t^2$ | 49.04 | 0.03 | 0.03 |
| $\max_{10 \leq t \leq 100} \tilde{B}_t^2$ | 1.26 | 1.34 | 1.50 |

Table 4 states maximums of $\tilde{G}_t^2$ and $\tilde{B}_t^2$ for the experiments of Section 4. This provides some theoretical explanation for empirical observations made during the experiments.

- **Smile Classification (4.1)** As discussed, PARALLEL TRAINING is at a disadvantage when $G \gg 0$ or $B \gg 1$, and Table 4 (left column) shows that $\tilde{G}_t$ is significantly large in smile classification. This aligns with our empirical observation in Figure 2a of PARALLEL TRAINING converging slower than either GRADIENT TRANSFER algorithm.
- **Language Modeling (4.2)** Table 4 (center column) shows language modeling with more trivial gradient dissimilarity: $\tilde{G}_t \approx 0$ and $\tilde{B}_t \approx 1$. PARALLEL TRAINING should be competitive. Figure 2b empirically confirms this to be true; all algorithms converge roughly equivalently.

## 6 CONCLUSION

This paper has introduced mixed FL, including motivation, algorithms and their convergence properties, and intuition for when a given algorithm will be useful for a given problem. Our experiments indicate mixed FL can improve accuracy and reduce communication and computation across tasks.

This work focused on jointly learning from a single decentralized client population and a centralized entity, as it illuminates the key aspects of the mixed FL problem. Note that mixed FL and the associated properties we define in this paper (like mixed FL $(G, B)$-BGD) are easily expanded to work with multiple $(> 1)$ distinct client populations participating. E.g., a population of mobile phones and a separate population of smart speakers, or mobile phones separated into populations with distinct capabilities/usage (high-end vs. low-end, or by country/language). Also, there need not be a centralized entity; mixing can be solely between distinct federated datasets.

It is interesting to reflect on the bounds of Table 3, and what they indicate about the benefits of separating a single decentralized client population into multiple populations for mixed FL purposes. The bounds are in terms of $\sigma^2$ (representing within population 'variability') and $G$ and $B$ (representing cross-population 'variability'). Splitting a population based on traits will likely decrease $\sigma^2$ (each population is now more homogeneous) but introduce or increase $G$ and $B$ (populations are now distinctive). This might indicate scenarios where GRADIENT TRANSFER methods (only bounded by $\sigma^2$) become more useful and PARALLEL TRAINING (also bound by $G$ and $B$) becomes less useful.

The limits of our convergence bounds should be noted. First, they are 'loose'; practical performance in particular algorithmic scenarios could be better, and thus comparisons between algorithms could differ. Second, our bounds assume IID federated data, which is invalid in practice; convergence properties differ on non-IID data. While our analysis, extended to handle non-IID data, shows that the bounds do not materially change, it is still a place where theory and practice slightly diverge.

It is important to note that mixed FL is *orthogonal* to the choice of algorithm for federated training. For simplicity, this paper described mixed FL with FEDAVG used for federated training. However, one could use DP-FedAvg (McMahan et al. (2018)) or DP-FTRL (Kairouz et al. (2021b)) with mixed FL to ensure that DP protections are applied to the federated data. Similarly, an adaptive SERVEROPTIMIZER like FEDADAM (Reddi et al., 2020) could be used in the federated portion of training (preliminary mixed FL results with FEDADAM are given in Appendix B.4.4).

In principle, mixed FL techniques are expected to have positive societal impacts insofar as they further develop the toolkit for FL (which has security and privacy benefits to users) and improve accuracy on final inference distributions. Also, we've shown (Section 4.2) how mixed FL can address participation biases that arise in FL. However, the addition of server-based data to federated optimization raises the possibility that biases in large public corpora find their way into more applications of FL.

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

# A PRACTICAL IMPLEMENTATION DETAILS

## A.1 DOWNLOAD SIZE

GRADIENT TRANSFER (either 1-WAY or 2-WAY) requires sending additional data as part of the communication from server to clients at the start of a federated round. Apart from the usual model checkpoint weights, with GRADIENT TRANSFER we must now also transmit 'augmenting' gradients of the model weights w.r.t centralized data as well. Naively, this doubles the download size as the gradient is the same size as the model. However, the augmenting gradients should be amenable to compression, e.g. using an approach such as Mitchell et al. (2022).

## A.2 UPLOAD SIZE

With PARALLEL TRAINING and 1-WAY GRADIENT TRANSFER, no client gradient information is used outside of the clients themselves, so there is no additional information (apart from the model deltas and aggregation weights) to upload to the server. With 2-WAY GRADIENT TRANSFER, client gradient information is used in centralized training, so needs to be conveyed to the server somehow.

When the FL client optimization is SGD, the average client gradient in a round (over all clients participating, over all steps) can be determined from the model deltas and aggregation weights that are already being sent back to the server, meaning no additional upload bandwidth is necessary. Apart from bandwidth considerations, this also means there is no additional vector for private data leakage. The algorithm to do this is as follows.

Each client $i$ transmits to the server a local model change $\Delta_i^{(t)}$ and an aggregation weight $p_i$ that is related to number of steps taken $K_i$. The average *total* gradient applied at client $i$ during round $t$ is:

$$\bar{g}^{(t)} = -\frac{1}{\eta K_i}\Delta_i^{(t)} \tag{8}$$

The average *client* gradient (i.e., w.r.t. just client data) at client $i$ is:

$$\bar{g}_i^{(t)} = -\frac{1}{\eta K_i}\Delta_i^{(t)} - \tilde{g}_c^{(t)} \tag{9}$$

where $\tilde{g}_c^{(t)}$ is the augmenting centralized gradient that was calculated from centralized data and used in round $t$. The average (across the cohort) of average client gradients, weighted by $K_i$, is:

$$\bar{g}_f^{(t)} = -\frac{1}{\eta \sum_i K_i}\sum_i \Delta_i^{(t)} - \tilde{g}_c^{(t)} \tag{10}$$

This average client gradient $\bar{g}_f^{(t)}$ is in the spirit of SCAFFOLD (Karimireddy et al., 2020b) Equation 4, Option II. It will be used as the augmenting federated gradient $\tilde{g}_f^{(t+1)}$ in the subsequent round $t + 1$, to augment centralized optimization. See Algorithm 1.

## A.3 DEBUGGING AND HYPERPARAMETER INTUITION VIA $K = 1$

As these algorithms each involve different hyperparameters, validating that software implementations[13] are behaving as expected is non-trivial. Something that proved useful for debugging purposes, as well as provided practical experience in understanding equivalences between the algorithms, was to perform test cases with the number of local steps $K$ set to 1. In this setting, the three mixed FL algorithms are effectively identical and should make equivalent progress during training.

Note that the convergence bounds of Table 3 hold for $K \geq 2$, so this takes us outside the operating regime where the bounds predict performance. It also takes us outside an operating regime that is typically useful (FL use cases generally find multiple steps per round to be beneficial). But it does serve a purpose when debugging.

---

[13] An open-source mixed FL code repository will be shared with the public version of the paper.

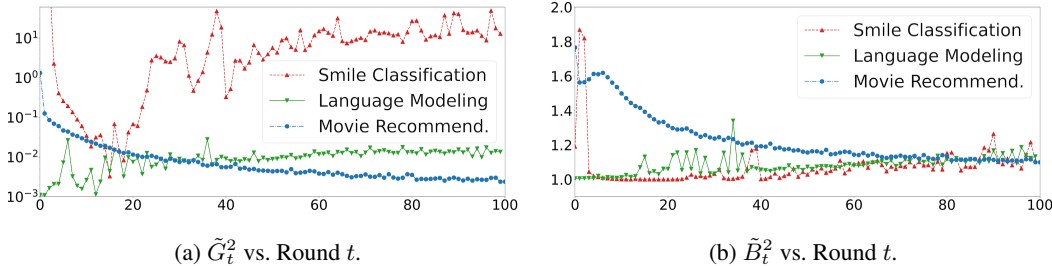

(a) $\tilde{G}_t^2$ vs. Round $t$.    (b) $\tilde{B}_t^2$ vs. Round $t$.

Figure 4: Sampled approximations of mixed FL $(G, B)$-BGD, for the three experiments in Section 4.

# B    EXPERIMENTS: ADDITIONAL INFORMATION AND RESULTS

## B.1    $\tilde{G}_t$ AND $\tilde{B}_t$ METRICS PLOTS

Table 4 is an informative comparison of the mixed FL optimization landscape of the three respective experiments conducted in Section 4. It includes maximum values for the metrics $\tilde{G}_t$ and $\tilde{B}_t$ (the sampled approximations of the parameters defined in $(G, B)$-BGD (Definition 5.2). Here we provide some additional information.

Figure 4 plots these sampled approximation metrics over the first 100 rounds of training. We ran 5 simulations per experiment and took the maximum at each round across simulations. We used the same hyperparameters as described below (in Subsection B.2), except taking only a single step per round ($K = 1$).

## B.2    ADDITIONAL DETAILS FOR EXPERIMENTS IN SECTION 4

**General Notes on Hyperparameter Selection**    For the various experiments in Section 4, we empirically determined good hyperparameter settings (as documented in Tables 5-10). Our general approach for each task was to leave server learning rate $\eta_s$ at 1, select a number of steps $K$ that made the most use of the examples in each client's cache, and then do a sweep of client learning rates $\eta$ to determine a setting that was fast but didn't diverge. For PARALLEL TRAINING and 2-WAY GRADIENT TRANSFER, which involve central optimization and merging, we set the merging learning rate $\eta_m$ to be 1, and set the central learning rate $\eta_c$ as the product of client and server learning rates: $\eta_c = \eta\eta_s$ (and since $\eta_s = 1$, this meant client and central learning rates were equal).

**General Notes on Comparing Algorithms**    We generally kept hyperparameters equivalent when comparing the algorithms. For example, we aimed to set batch sizes for all algorithms such that central and client gradient variances $\sigma^2$ and $\sigma_c^2$ have equivalent impact on convergence (meaning $|\mathcal{B}_c| = S|\mathcal{B}_i|$ for PT and 2-w GT, and $|\mathcal{B}_c| = KS|\mathcal{B}_i|$ for 1-w GT). In the case of language model training with 1-WAY GRADIENT TRANSFER, following this rubric would have meant a central batch size $|\mathcal{B}_c|$ of 12800; we reduced this in half for practical computation reasons. For a given task, we also generally kept learning rates the same for all algorithms. Interestingly, we observed that as $\eta$ (and $\eta_c$, if applicable) is increased for a given task, the 2-WAY GRADIENT TRANSFER algorithm is the first of the three to diverge, and so we had to adjust, e.g., in the language modeling experiment we used a lower $\eta$ for 2-w GT than for PT and 1-w GT.

### B.2.1    CELEBA SMILE CLASSIFICATION

**Datasets**    The CelebA federated dataset consists of 9,343 raw clients, which can be broken into train/evaluation splits of 8,408/935 clients, respectively (TFF CelebA documentation, 2022). The raw clients have average cache size of $\sim 21$ face images. The images are about equally split between smiling and unsmiling faces. In order to enlarge cache size, we group three raw clients together into one composite client, so our federated training data involves 2,802 clients with caches of (on average) $\sim 63$ face images (and about half that when we limit the clients to only have smiling faces).

Table 5: Smile classifier training, federated hyperparameters.

| | | (all) | | |
|---|---|---|---|---|
| $S$ | $|\mathcal{B}_i|$ | $K$ | $\eta$ | $\eta_{\mathrm{s}}$ |
| 100 | 5 | 2 | 0.01 | 1.0 |

Table 6: Smile classifier training, centralized and overall hyperparameters.

| (1-w GT) | (PT and 2-w GT) | | | | (all) | |
|---|---|---|---|---|---|---|
| $|\mathcal{B}_c|$ | $|\mathcal{B}_c|$ | $K$ | $\eta_{\mathrm{c}}$ | $\eta_{\mathrm{m}}$ | $w_{\mathrm{f}}$ | $w_{\mathrm{c}}$ |
| 1000 | 500 | 2 | $=\eta$ | 1.0 | 0.5 | 0.5 |

Table 7: Language model training, federated hyperparameters.

| | | (all) | | |
|---|---|---|---|---|
| $S$ | $|\mathcal{B}_i|$ | $K$ | $\eta$ | $\eta_{\mathrm{s}}$ |
| 100 | 8 | 16 | 2.0 (2-w GT: 1.0) | 1.0 |

Table 8: Language model training, centralized and overall hyperparameters.

| (1-w GT) | (PT and 2-w GT) | | | | (all) | |
|---|---|---|---|---|---|---|
| $|\mathcal{B}_c|$ | $|\mathcal{B}_c|$ | $K$ | $\eta_{\mathrm{c}}$ | $\eta_{\mathrm{m}}$ | $w_{\mathrm{f}}$ | $w_{\mathrm{c}}$ |
| 6400 | 800 | 16 | $=\eta$ | 1.0 | 0.75 | 0.25 |

Table 9: Movie recommender training, federated hyperparameters.

| | | (all) | | |
|---|---|---|---|---|
| $S$ | $|\mathcal{B}_i|$ | $K$ | $\eta$ | $\eta_{\mathrm{s}}$ |
| 100 | 16 | 10 | 0.5 | 1.0 |

Table 10: Movie recommender training, centralized and overall hyperparameters.

| (1-w GT) | (PT and 2-w GT) | | | | (all) | |
|---|---|---|---|---|---|---|
| $|\mathcal{B}_c|$ | $|\mathcal{B}_c|$ | $K$ | $\eta_{\mathrm{c}}$ | $\eta_{\mathrm{m}}$ | $w_{\mathrm{f}}$ | $w_{\mathrm{c}}$ |
| − | − | 10 | $=\eta$ | 1.0 | 0.5 | 0.5 |

Our evaluation data consists of both smiling and unsmiling faces, and is meant to stand in for the inference distribution (where accurate classification of both smiling and unsmiling inputs is necessary). Note that as CelebA contains smiling and unsmiling faces in nearly equal amounts, a high evaluation accuracy cannot come at the expense of one particular label being poorly classified.

**Model Architecture** The architecture used is a very basic fully-connected neural network[14] with a single hidden layer of 64 neurons with ReLU activations.

**Hyperparameter Settings** The settings used in mixed FL training are shown in Tables 5 and 6. SGD was used for all optimizers: CLIENTOPTIMIZER and SERVEROPTIMIZER, and (if PT or 2-w GT) CENTRALOPTIMIZER and MERGEOPTIMIZER.

### B.2.2 STACK OVERFLOW/WIKIPEDIA LANGUAGE MODELING

**Datasets** The Stack Overflow dataset is a large-scale federated dataset, consisting of 342,477 training clients and 204,088 evaluation clients (TFF StackOverflow documentation, 2022). The training clients have average cache size of $\sim 400$ examples, and evaluation clients have average cache size of $\sim 80$ examples. The Wikipedia dataset (`wikipedia/20201201.en`) consists of 6,210,110 examples (TFDS Wikipedia (20201201.en) documentation, 2022), which we partition by designating the first 5,000,000 examples for training and the remainder for evaluation. All raw text data is processed into sequences of 100 characters.

Our evaluation data is a combined dataset consisting of randomly shuffled examples drawn from the Stack Overflow evaluation clients and the Wikipedia dataset. After accounting for raw examples discarded during processing, the evaluation dataset is comprised of approximately 75% Stack Overflow and 25% Wikipedia. Hence, we used this weighting proportion when performing mixed FL training in the language modeling problem (Table 8).

---

[14]Adapted from an online tutorial involving CelebA binary attribute classification: *"TensorFlow Constrained Optimization Example Using CelebA Dataset"*.

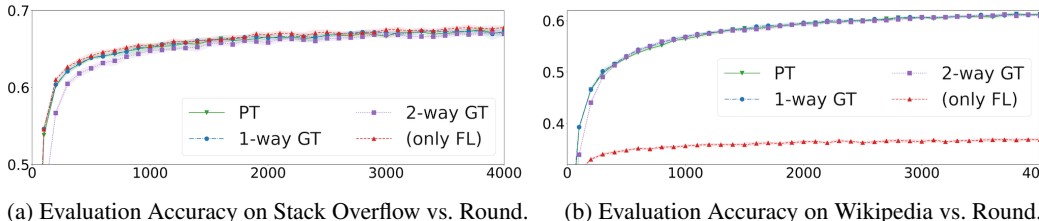

(a) Evaluation Accuracy on Stack Overflow vs. Round.   (b) Evaluation Accuracy on Wikipedia vs. Round.

Figure 5: Language model accuracy, evaluated on only decentralized (left) or centralized (right) data.

**Model Architecture**   The architecture used is a recurrent neural network (RNN)[15] with an embedding dimension of 256 and 1024 GRU units.

**Hyperparameter Settings**   The settings used in mixed FL training are shown in Tables 7 and 8. Note that in this experiment we also found clipping the norm of gradients (both federated and centralized) to be beneficial, for all mixed FL algorithms. We used adaptive clipping for the federated gradients (via the method described in Andrew et al. (2021), without noise addition) and a fixed clip value of 0.2 for the centralized gradients. SGD was used for all optimizers: CLIENTOPTIMIZER and SERVEROPTIMIZER, and (if PT or 2-W GT) CENTRALOPTIMIZER and MERGEOPTIMIZER.

**Evaluation Accuracy on Individual Data Splits**   Figure 5 shows the accuracy of models (trained either via mixed FL or pure FL) when evaluated on only federated data (Stack Overflow) or only centralized data (Wikipedia) individually. Confirming what we expect, the various mixed FL algorithms do a good job of achieving accuracy on both datasets. But if we train only via FL (without mixing), then we do a good job of learning the federated data (Stack Overflow) character sequences, but aren't nearly as accurate at predicting the next character in centralized data (Wikipedia) sequences.

### B.2.3   MOVIELENS MOVIE RECOMMENDATION

**Dataset**   The MovieLens 1M dataset (GroupLens) contains approximately 1 million ratings from 6,040 users on 3,952 movies. Examples are grouped by user, forming a natural data partitioning across clients. For all mixed FL algorithms we study, we keep examples from 20% of users (randomly selecting 20% of users and shuffling the examples of these users) as the datacenter data, and use examples from the remaining 80% users as client data. With this data splitting strategy, the server data won't include the same individual client distributions but it will still be sampled from the same meta distribution of clients. We then split the clients data into the train and test sets, resulting in 3,865 train, 483 validation, and 483 test users. The average cache size of each client is $\sim 160$ examples.

**Model Architecture**   The architecture used is the same as Anonymous (b), a dual encoder representation learning model with a bag-of-word encoder for the left tower (which takes in a list of movies a user has seen) and a simple embedding lookup encoder for the right tower (which takes in the next movie a user sees).

**Hyperparameter Settings**   The settings used in mixed FL training are shown in Tables 9 and 10. In Subsection 4.3, SGD was used for all optimizers: CLIENTOPTIMIZER and SERVEROPTIMIZER, and (if PT or 2-W GT) CENTRALOPTIMIZER and MERGEOPTIMIZER. See Appendix B.4.4 for an additional experiment where ADAM is used as the SERVEROPTIMIZER in 1-W GT.

**Recall@10**   As mentioned in Subsection 4.3, all mixed FL algorithms achieved similar global recall@10 compared to the baseline. Figure 6 shows evaluation recall@10 over 2000 training rounds.

---

[15]Adapted from an online tutorial involving next character prediction: *"Text generation with an RNN"*.

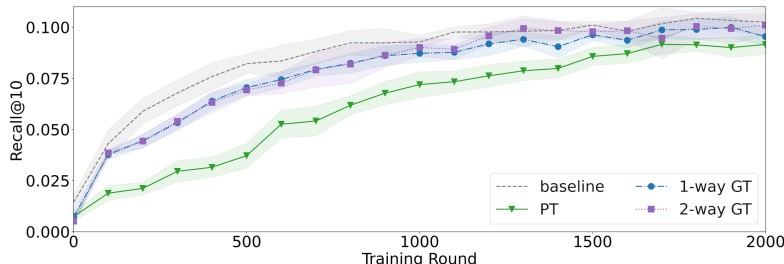

Figure 6: Next movie prediction performance (recall@10).

Table 11: Computation and communication overheads for Movie Recommendation task.

|  | baseline | PARALLEL TRAINING | 1-w GT | 2-w GT |
|---|---|---|---|---|
| Comp. | $(K \cdot N^2 \cdot d)/2$ | $(N^2 \cdot d)/2$ | $(N^2 \cdot d)/2$ | $(N^2 \cdot d)/2$ |
| Comm. | $2 \cdot N \cdot d$ | $2 \cdot n \cdot d$ | $3 \cdot n \cdot d$ | $3 \cdot n \cdot d$ |

### B.3 COMPUTATION AND COMMUNICATION SAVINGS FOR MOVIE RECOMMENDATION

This section provides a detailed analysis of the computation and communication savings brought by mixed FL in the movie recommendation task.

For movie recommendation, both the input feature and the label are movie IDs with a vocabulary size of $N$. They share the same embedding table, with an embedding dimension of $d$. The input features and label embedding layers account for most of the model parameters in a dual encoder. Therefore, we use the total size of the feature and label embeddings to approximate the model size: $M = (N + N) \cdot d$. Batch size is $\mathcal{B}_i$ and local steps per round is $K$. Let the averaged number of movies in each client's local dataset for each training round be $n$, smaller than $\mathcal{B}_i \cdot K$.

**Computation** As shown in the second row of Table 11, the amount of computation for regularization term is $(K \cdot N^2 \cdot d)/2$ if calculating on-device (baseline). When computing the regularization term on the server (mixed FL), the complexity is $(N^2 \cdot d)/2$. The total computation saving with mixed FL is $((K-1) \cdot N^2 \cdot d)/2$. We use $(N^2 \cdot d)/2$ instead of $N^2 \cdot d$ for regularization term computation which is more accurate for an optimized implementation.

The total computation complexity of the forward pass is $O(\mathcal{B}_i d + \mathcal{B}_i d^2 + \mathcal{B}_i^2 d)$, where the three items are for the bag-of-word encoder, the context hidden layer, and similarity calculation. The hinge loss and spreadout computation is $O(\mathcal{B}_i) + O(0.5N^2 d)$. The gradient computation is $O(2\mathcal{B}_i d^2 + 2\mathcal{B}_i^2 d)$ for network backward pass and $O(\mathcal{B}_i) + O(Nd)$ for hinge and spreadout. Therefore, when computing the regularization term on the server with mixed FL, the computation savings for each client is $1 - (\mathcal{B}_i d + 3\mathcal{B}_i d^2 + 3\mathcal{B}_i^2 d + 2\mathcal{B}_i)/(\mathcal{B}_i d + 3\mathcal{B}_i d^2 + 3\mathcal{B}_i^2 d + 2\mathcal{B}_i + 0.5N^2 d + Nd)$, which is 99.98% for all mixed FL algorithms.

**Communication** The communication overheads of each algorithm are presented in the last row of Table 11. For the baseline, the server and each client need to communicate the full embedding table and the gradients, so the communication overhead is $2 \cdot N \cdot d$ or 494KB. With PARALLEL TRAINING, the server and each client only communicate movie embeddings and the gradients corresponding to movies in that client's local datasets. Thus the communication traffic is reduced to $2 \cdot n \cdot d$ or 20KB. GRADIENT TRANSFER requires the server to send both the movie embeddings and gradients to each client. The communication overhead then becomes $3 \cdot n \cdot d$ or 30KB. Overall, mixed FL can save more than 93.9% communication overhead than the baseline.

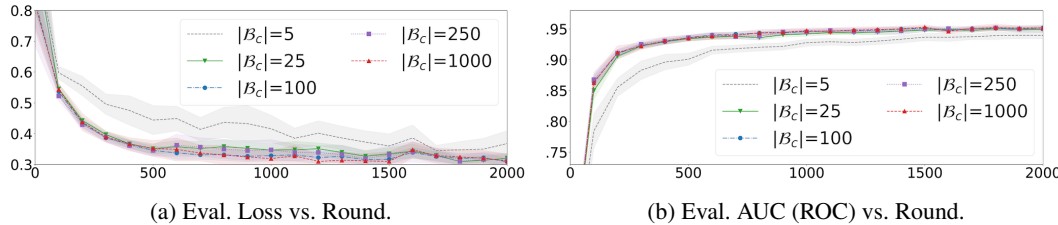

(a) Eval. Loss vs. Round.       (b) Eval. AUC (ROC) vs. Round.

Figure 7: Smile classifier training, 1-w GT with various central batch sizes $|\mathcal{B}_{\mathrm{c}}|$.

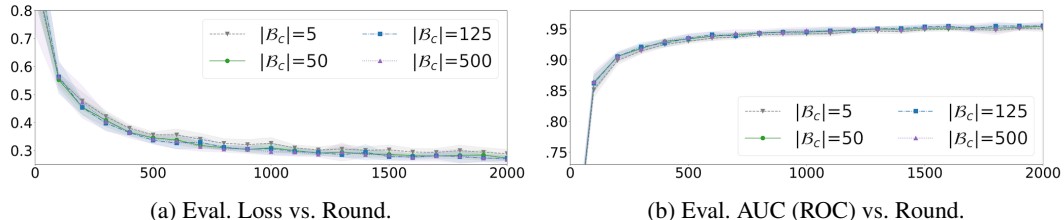

(a) Eval. Loss vs. Round.       (b) Eval. AUC (ROC) vs. Round.

Figure 8: Smile classifier training, 2-w GT with various central batch sizes $|\mathcal{B}_{\mathrm{c}}|$.

### B.4 ADDITIONAL OBSERVATIONS AND EXPERIMENTS

#### B.4.1 EFFECT OF $\sigma_{\mathrm{c}}^2$ ON CONVERGENCE

Table 3 shows that the theoretical bounds on rounds to convergence are directly proportional to the client variance bound $\sigma^2$ and central variance bound $\sigma_{\mathrm{c}}^2$. Also, as discussed in Subsection 5.2, 1-WAY GRADIENT TRANSFER is more sensitive to high central variance than the other two algorithms. Whereas in the other algorithms the impact of $\sigma_{\mathrm{c}}^2$ on convergence scales with cohort size $S$, in 1-WAY GRADIENT TRANSFER it scales with cohort size $S$ *and* steps taken per round $K$.

To observe the effect of $\sigma_{\mathrm{c}}^2$ in practice, and compare its effect on 1-WAY GRADIENT TRANSFER vs. 2-WAY GRADIENT TRANSFER, we ran sweeps of CelebA smile classification training, varying the central batch size $|\mathcal{B}_{\mathrm{c}}|$. The plots of evaluation loss and evaluation AUC of ROC are shown in Figures 7 (1-w GT) and 8 (2-w GT). For each central batch size setting, we ran 10 trials; the plots show the means of each setting's trials, with corresponding 95% confidence bounds.

Figure 7 confirms the sensitivity of 1-WAY GRADIENT TRANSFER to central variance, with experiments using larger central batches $\mathcal{B}_{\mathrm{c}}$ converging faster than experiments using smaller central batches. However, at least in the case of this task, the benefits of lower variance disappear quickly. The convergence of AUC of ROC did not appreciably improve for central batch sizes larger than 25. Presumably there is little effect at these larger central batch sizes because in these cases the convergence is now dominated by *client* variance (i.e., further convergence improvements would come from increasing client batch size $|\mathcal{B}_i|$).

Comparing Figure 8 with Figure 7, we empirically observe that 2-WAY GRADIENT TRANSFER has lower sensitivity than 1-WAY GRADIENT TRANSFER to central batch size/central variance.

#### B.4.2 TRADING $\eta$ FOR $K$

The convergence bounds of Table 3 have an additional implication, in regards to the trade off between client learning rate $\eta$ (and central learning rate $\eta_{\mathrm{c}}$) and number of local steps taken $K$.

**It's better to reduce $\eta$ and $\eta_{\mathbf{c}}$ and increase $K$, but there are limits** The convergence bounds are not related to client or central learning rate ($\eta$ or $\eta_{\mathrm{c}}$), but are inversely related to local steps $K$. In general, it's best to take as many steps as possible, and if necessary reduce learning rates accordingly. But there are limits to how large $K$ can be. First, clients have finite caches of data, and $K$ will always be limited by cache size divided by batch size. Second, in the case of 1-WAY GRADIENT TRANSFER,

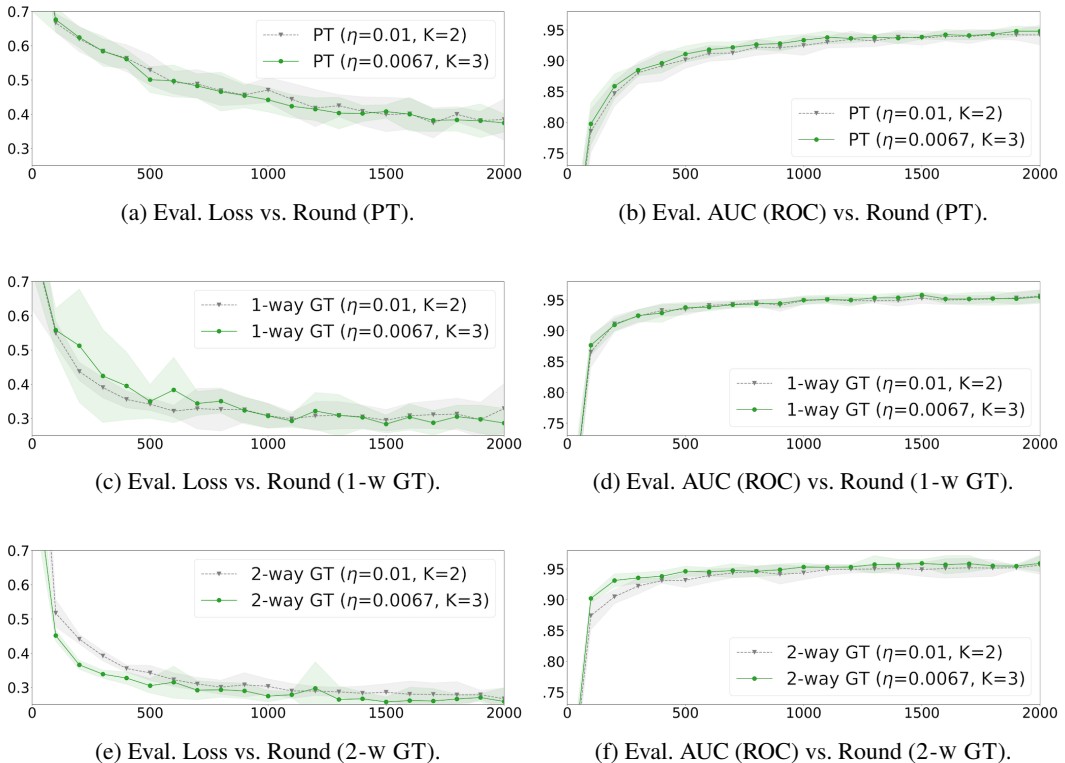

Figure 9: Smile classifier training with different $\eta$ and $K$ settings (for each mixed FL algorithm).

any increase in $K$ means that central variance $\sigma_c^2$ must be proportionally reduced (as mentioned above), necessitating even larger central batch sizes (which at some point is infeasible).

We empirically observed this relationship by running smile classification (Figure 9) and language modeling (Figure 10) experiments where client learning rate $\eta$ (and central learning rate $\eta_c$) are inversely proportionally varied with $K$. For each hyperparameter configuration we ran 5 trials; the figures include 95% confidence intervals. The results confirm that reducing these learning rates, and making a corresponding increase in the number of steps, is beneficial. It never hurts convergence, and often helps.

### B.4.3  DIFFERENCES IN EFFECTIVE STEP SIZE

Table 12 in Appendix C shows that in order to yield the convergence bounds stated in this paper, each algorithm makes different assumptions of maximum effective step size. From this we draw one final implication in regards to comparing the mixed FL algorithms.

**For given $\eta$, maximum $K$ varies by algorithm, *or*, for given $K$, maximum $\eta$ varies by algorithm** Consider just effective federated step size $\tilde{\eta} = \eta \eta_s K$ for the moment. Assume that server learning rate $\eta_s$ is held constant. Then each mixed FL algorithm has a different theoretical upper bound on the product of client learning rate $\eta$ and local steps per round $K$. If using a common $\eta$, the theoretical upper limit on $K$ varies by mixed FL algorithm. Alternatively if using a common $K$, the theoretical upper limit on $\eta$ varies by mixed FL algorithm.

The maximum effective step sizes of Table 12 imply that 2-WAY GRADIENT TRANSFER has narrower limits than 1-WAY GRADIENT TRANSFER on the allowable ranges of $\eta$ and $K$. It also indicates that for PARALLEL TRAINING the allowable range of $\eta$, $\eta_c$, and $K$ depends on the $B$ parameter from mixed FL $(G, B)$-BGD (Definition 5.2).

Some of this behavior has been observed empirically, when hyperparameter tuning our experiments (discussed in Subsection B.2). For example, for the language modeling experiment, assuming a

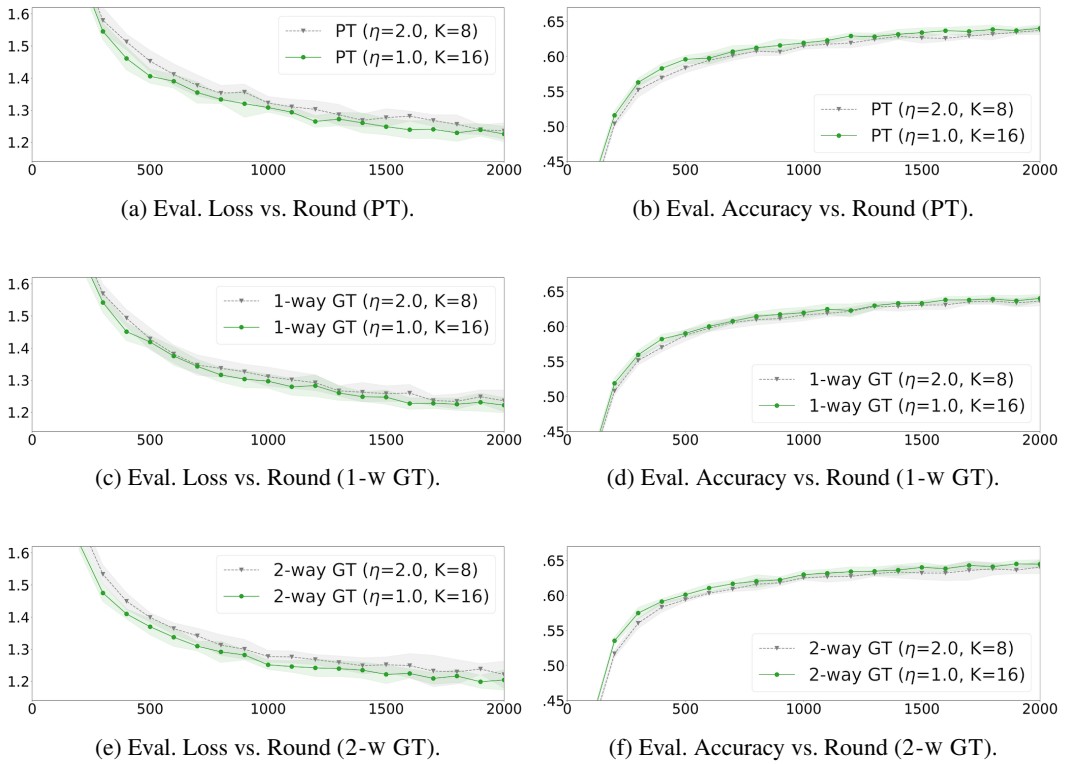

(a) Eval. Loss vs. Round (PT).    (b) Eval. Accuracy vs. Round (PT).

(c) Eval. Loss vs. Round (1-w GT).    (d) Eval. Accuracy vs. Round (1-w GT).

(e) Eval. Loss vs. Round (2-w GT).    (f) Eval. Accuracy vs. Round (2-w GT).

Figure 10: Language model training with different $\eta$ and $K$ settings (for each mixed FL algorithm).

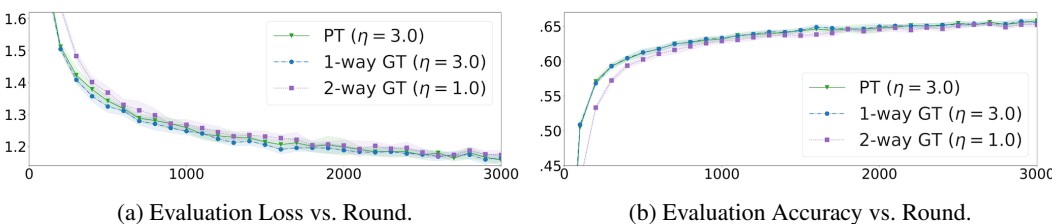

(a) Evaluation Loss vs. Round.    (b) Evaluation Accuracy vs. Round.

Figure 11: Language model training, with higher $\eta$ for PT and 1-w GT ($K = 16$ for all). The increased learning rate boosts progress on evaluation loss and accuracy early in optimization, but does not change the number of rounds ultimately required for convergence. All three algorithms have reached similar loss and accuracy by round 3000, and are still converging.

constant number of steps of $K = 16$, 2-WAY GRADIENT TRANSFER tends to diverge when learning rate $\eta$ was increased beyond 1.0, whereas 1-WAY GRADIENT TRANSFER is observed to converge even with learning rate $\eta$ of 5.0. (PARALLEL TRAINING is in-between; it still converges with learning rate $\eta$ of 3.0, but diverges when learning rate $\eta$ is 5.0.) An interesting characteristic to note is that using different $\eta$ in different algorithms does not really impact comparative convergence. Figure 11 shows convergence in the language modeling experiment, when 2-WAY GRADIENT TRANSFER uses $\eta = 1.0$ and 1-WAY GRADIENT TRANSFER and PARALLEL TRAINING both use $\eta = 3.0$ (in all cases, with $K = 16$). The higher learning rate of 1-W GT and PT helps a little early, but does not impact the number of rounds to convergence. This holds with the theoretical convergence bounds of Table 3, which show a relationship with steps $K$ but not learning rates (as also discussed above).

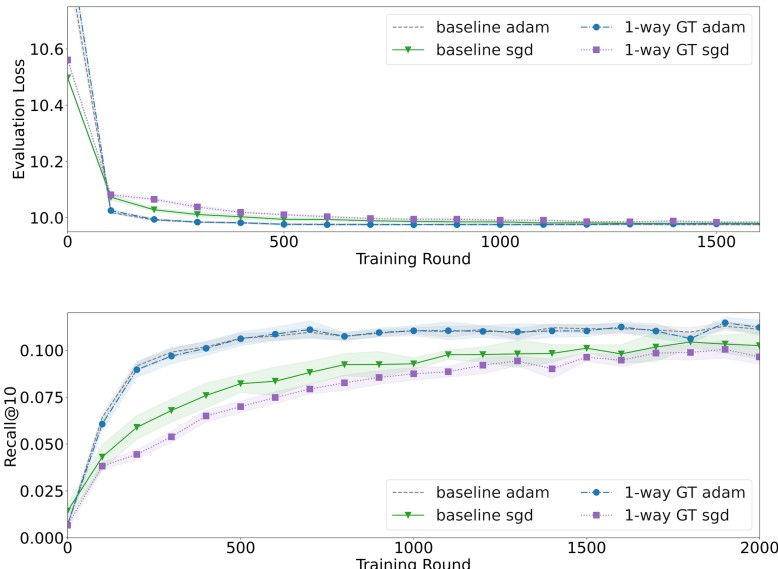

Figure 12: Next movie prediction performance when training with adaptive optimizer (client optimizer: SGD, server optimizer: ADAM).

### B.4.4 1-W GT WITH ADAPTIVE OPTIMIZATION

We briefly studied the performance of 1-WAY GRADIENT TRANSFER when using ADAM in place of SGD as the server optimizer, i.e., FEDADAM (Reddi et al., 2020). Note that the server adaptive optimizer requires a smaller learning rate to perform well. Figure 12 reports the results of using ADAM as the server optimizer with a server learning rate of 0.01. All the other hyperparameters are the same as in Tables 9 and 10. We observe that (1) ADAM works better than SGD, leading to better convergence, and (2) 1-WAY GRADIENT TRANSFER performs almost the same as the baseline when using ADAM. We will extend our investigation of mixed FL with adaptive optimization in the future. This will include studying methods for applying adaptive optimization to PARALLEL TRAINING and 2-WAY GRADIENT TRANSFER; these algorithms are more complicated since they involve additional optimizers (CENTRALOPTIMIZER and MERGEOPTIMIZER).

### B.4.5 COMPARISON WITH TRANSFER LEARNING

Section 3 discussed how transfer learning is ill-suited for mixed FL. Here we show why.

Recall the bias mitigation mixing task of Section 4.2. The end goal is a single trained language model that performs well both for high-end *and* low-end mobile phone users. We use Stack Overflow and Wikipedia datasets as respective experimental stand-ins, as they model the problem well. They are in the same domain (Latin alphabet) and same language (English) but with distinct usage distributions.

Transfer learning sequentially trains the language model, first 'pretraining' on one dataset and then switching to 'fine-tune' on the other. One could first centrally pretrain (with Wikipedia), then switch to federated fine-tuning (with Stack Overflow), or do the converse. Figure 13 shows what happens. The switches take place at 250 rounds. We also show PARALLEL TRAINING as a point of comparison.

Figure 13 shows that in transfer learning, catastrophic forgetting of the pretraining task occurs after switching to the fine-tuning task. The centralized pretraining case precipitously drops its evaluation accuracy on datacenter data (Wikipedia). The federated pretraining case does the same with federated data (Stack Overflow). Figure 13a shows that PARALLEL TRAINING achieves superior evaluation accuracy on the stand-in for the desired inference distribution (i.e. the combined Stack Overflow and Wikipedia datasets).

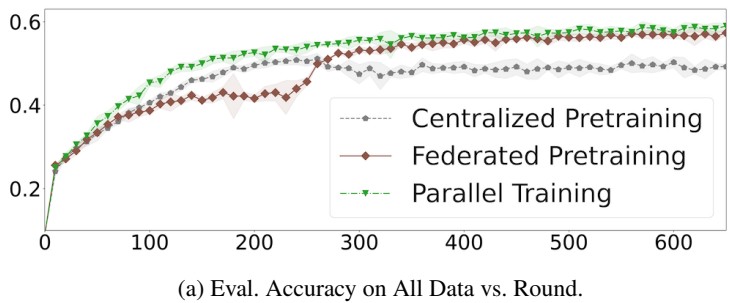

(a) Eval. Accuracy on All Data vs. Round.

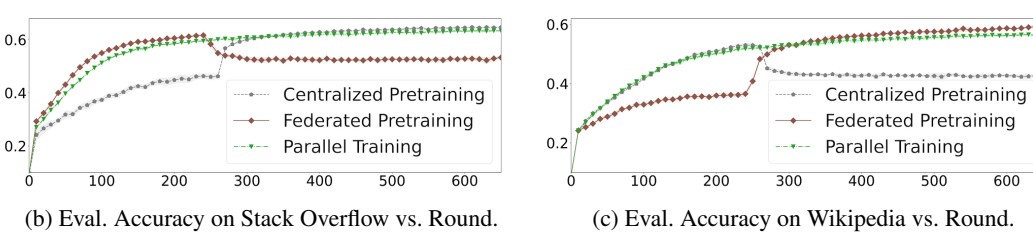

(b) Eval. Accuracy on Stack Overflow vs. Round.    (c) Eval. Accuracy on Wikipedia vs. Round.

Figure 13: Mixed FL endeavors to learn *both* data distributions, but transfer learning fails to retain knowledge of the pretraining task. When the pretraining-to-fine-tuning switch happens at round 250, the transfer learning cases fit to the new distribution and 'forget' the previous distribution they were pretrained on. PARALLEL TRAINING does not suffer from this problem as it sees data from both distributions throughout training.

To simplify Figure 13 we don't show GRADIENT TRANSFER results, but they would match PARALLEL TRAINING (e.g., as shown in Figures 1b and 2b). Because the PARALLEL TRAINING and GRADIENT TRANSFER algorithms presented in this paper jointly train on both data distributions in parallel (as opposed to sequentially), they avoid catastrophic forgetting.

Table 12: Maximum effective federated step size, $\tilde{\eta} = \eta\eta_s K$, for convergence bounds in Appendix C and Table 3. When applicable (PT, 2-w GT) the effective centralized step size, $\eta_c K$, shares the same maximum (and assume that merging learning rate $\eta_m$ is 1). $\beta$ is the smoothness bound (Def. D.1).

|  | PT | 1-w GT | 2-w GT |
|---|---|---|---|
| $\mu$-CONVEX | $\frac{1}{6(1+B^2)\beta}$ | $\frac{1}{8\beta}$ | $\min\left(\frac{1}{81\beta}, \frac{1}{15\mu}\right)$ |
| CONVEX | $\frac{1}{6(1+B^2)\beta}$ | $\frac{1}{8\beta}$ | $\frac{1}{81\beta}$ |
| NONCONVEX | $\frac{1}{6(1+B^2)\beta}$ | $\frac{1}{18\beta}$ | $\frac{1}{24\beta}$ |

Table 13: Assumptions on merging or server learning rates, for convergence bounds in Appendix C and Table 3.

|  | PT | 1-w GT | 2-w GT |
|---|---|---|---|
| (ASSUMES) | $\eta_m \geq 1$ | $\eta_s \geq \sqrt{S}$ | $\eta_m \geq 1$ |

# C CONVERGENCE THEOREMS

The three subsections that follow state theorems for convergence (to an error smaller than $\epsilon$) for the respective mixed FL algorithms. The convergence bounds are summarized in Table 3 in Section 5. Tables 12 and 13 convey some supporting aspects of the convergence bounds, about limits on effective step size ($\tilde{\eta} = \eta\eta_s K$) and assumptions on learning rates.

## C.1 PARALLEL TRAINING

Given Assumption 5.1, one can view PARALLEL TRAINING as a 'meta-FEDAVG' involving two 'meta-clients'. One meta-client is the population of IID federated clients (collectively having loss $f_f$), and the other meta-client is the centralized data at the datacenter (having loss $f_c$). As such, we can take the convergence theorem for FEDAVG derived in Karimireddy et al. (2020b) (Section 3, Theorem I) and observe that it applies to the number of rounds $T$ to reach convergence in the PARALLEL TRAINING scenario.

**Theorem C.1.** *For PARALLEL TRAINING, where the federated data is IID (Assumption 5.1), for $\beta$-smooth functions $f_f$ and $f_c$ which satisfy Definition 5.2, the number of rounds $T$ to reach an expected error smaller than $\epsilon$ is:*

$$\mu\text{-Strongly convex:} \quad T = \tilde{\mathcal{O}}\left(\frac{(\sigma^2+S\sigma_c^2)}{KS\mu\epsilon} + \frac{G\sqrt{\beta}}{\mu\sqrt{\epsilon}} + \frac{B^2\beta}{\mu}\log(\frac{1}{\epsilon})\right)$$

$$\text{General convex:} \quad T = \mathcal{O}\left(\frac{(\sigma^2+S\sigma_c^2)D^2}{KS\epsilon^2} + \frac{G\sqrt{\beta}}{\epsilon^{\frac{3}{2}}} + \frac{B^2\beta D^2}{\epsilon}\right)$$

$$\text{Non-convex:} \quad T = \mathcal{O}\left(\frac{(\sigma^2+S\sigma_c^2)\beta F}{KS\epsilon^2} + \frac{G\sqrt{\beta}}{\epsilon^{\frac{3}{2}}} + \frac{B^2\beta F}{\epsilon}\right)$$

*where $F = f(\boldsymbol{x}^{(0)}) - f(\boldsymbol{x}^*)$, $D^2 = \left\|\boldsymbol{x}^{(0)} - \boldsymbol{x}^*\right\|^2$. Conditions for above: $\eta_m \geq 1; \eta_c, \eta\eta_s \leq \frac{1}{6(1+B^2)\beta K\eta_m}$.*

*Proof.* The analysis is exactly along the lines of the analysis in Karimireddy et al. (2020b), Appendix D.2, in the context of FEDAVG. Effectively, the analysis applies to the 'meta-FEDAVG' problem of PARALLEL TRAINING, with two 'meta-clients', one being the central loss/data (with stochastic gradients with variance of $\sigma_c^2$) and the other being the federated loss/data. The homogeneity of the clients and the averaging over the sampled clients effectively reduces the variance of the stochastic gradients to $\sigma^2/S$. The analysis follows in a straightforward manner by accounting for the variance in appropriate places. We omit the details for brevity. □

## C.2 1-WAY GRADIENT TRANSFER

We now provide convergence bounds for the 1-WAY GRADIENT TRANSFER scenario. Unlike PARALLEL TRAINING, which could be thought of as a 'meta' version of an existing FL algorithm (FEDAVG), 1-WAY GRADIENT TRANSFER is an entirely new FL algorithm. As such, we must formulate a novel proof (Appendix D) of its convergence bounds.

Given Assumption 5.1, the following Theorem gives the number of rounds to reach a given expected error.

**Theorem C.2.** *For* 1-WAY GRADIENT TRANSFER, *where the federated data is IID (Assumption 5.1), for $\beta$-smooth functions $f_i$ and $f_c$, the number of rounds $T$ to reach an expected error smaller than $\epsilon$ is:*

$$
\begin{aligned}
\text{$\mu$-Strongly convex:} \quad & T = \tilde{\mathcal{O}}\left( \frac{(\sigma^2 + KS\sigma_c^2)}{KS\mu\epsilon} + \frac{\beta}{\mu} \log(\tfrac{1}{\epsilon}) \right) \quad && \text{when } \eta_s > \sqrt{\tfrac{5}{8}S}, \eta \le \tfrac{1}{8\beta K\eta_s} \\
\text{General convex:} \quad & T = \mathcal{O}\left( \frac{(\sigma^2 + KS\sigma_c^2)D^2}{KS\epsilon^2} + \frac{\beta D^2}{\epsilon} \right) \quad && \text{when } \eta_s > \sqrt{\tfrac{5}{8}S}, \eta \le \tfrac{1}{8\beta K\eta_s} \\
\text{Non-convex:} \quad & T = \mathcal{O}\left( \frac{(\sigma^2 + KS\sigma_c^2)\beta F}{KS\epsilon^2} + \frac{\beta F}{\epsilon} \right) \quad && \text{when } \eta_s \ge \sqrt{S}, \eta \le \tfrac{1}{18\beta K\eta_s}
\end{aligned}
$$

*where $F = f(\boldsymbol{x}^{(0)}) - f(\boldsymbol{x}^*)$, $D^2 = \left\| \boldsymbol{x}^{(0)} - \boldsymbol{x}^* \right\|^2$.*

*Proof.* Detailed proof given in Appendix D. □

## C.3 2-WAY GRADIENT TRANSFER

Given Assumption 5.1, one can view 2-WAY GRADIENT TRANSFER as a 'meta-SCAFFOLD' involving two 'meta-clients' (analogous to the view of PARALLEL TRAINING as 'meta-FEDAVG' in Subsection C.1). As such, we can take the convergence theorem for SCAFFOLD derived in Karimireddy et al. (2020b) (Section 5, Theorem III) and observe that it applies to the number of rounds $T$ to reach convergence in the 2-WAY GRADIENT TRANSFER scenario.

**Theorem C.3.** *For* 2-WAY GRADIENT TRANSFER, *where the federated data is IID (Assumption 5.1), for $\beta$-smooth functions $f_f$ and $f_c$, the number of rounds $T$ to reach an expected error smaller than $\epsilon$ is:*

$$
\begin{aligned}
\text{$\mu$-Strongly convex:} \quad & T = \tilde{\mathcal{O}}\left( \frac{(\sigma^2 + S\sigma_c^2)}{KS\mu\epsilon} + \frac{\beta}{\mu} \log(\tfrac{1}{\epsilon}) \right) \quad && \text{when } \eta_m \ge 1; \eta_c, \eta\eta_s \le \min\left( \tfrac{1}{81\beta K\eta_m}, \tfrac{1}{15\mu K\eta_m} \right) \\
\text{General convex:} \quad & T = \mathcal{O}\left( \frac{(\sigma^2 + S\sigma_c^2)D^2}{KS\epsilon^2} + \frac{\beta D^2}{\epsilon} \right) \quad && \text{when } \eta_m \ge 1; \eta_c, \eta\eta_s \le \tfrac{1}{81\beta K\eta_m} \\
\text{Non-convex:} \quad & T = \mathcal{O}\left( \frac{(\sigma^2 + S\sigma_c^2)\beta F}{KS\epsilon^2} + \frac{\beta F}{\epsilon} \right) \quad && \text{when } \eta_m \ge 1; \eta_c, \eta\eta_s \le \tfrac{1}{24\beta K\eta_m}
\end{aligned}
$$

*where $F = f(\boldsymbol{x}^{(0)}) - f(\boldsymbol{x}^*)$, $D^2 = \left\| \boldsymbol{x}^{(0)} - \boldsymbol{x}^* \right\|^2$.*

*Proof.* The analysis is exactly along the lines of the analysis in Karimireddy et al. (2020b), Appendix E, in the context of SCAFFOLD. Effectively, the analysis applies to the 'meta-SCAFFOLD' problem of 2-WAY GRADIENT TRANSFER, with two 'meta-clients', one being the central loss/data (with stochastic gradients with variance of $\sigma_c^2$) and the other being the federated loss/data. The homogeneity of the clients and the averaging over the sampled clients effectively reduces the variance of the stochastic gradients to $\sigma^2/S$. The analysis follows in a straightforward manner by accounting for the variance in appropriate places. We omit the details for brevity. □

## D CONVERGENCE PROOFS FOR 1-WAY GRADIENT TRANSFER

We will prove the convergence rate of 1-WAY GRADIENT TRANSFER for 3 different cases: strongly convex, general convex, and non-convex. While these proofs are influenced by those for SCAFFOLD in Karimireddy et al. (2020b), we note some additional technical challenges we face with 1-WAY GRADIENT TRANSFER. First, as the name indicates, gradient information only flows one way (in SCAFFOLD, gradient information flows from and to all clients). Second, we only sample a batch of centralized data once per round (whereas SCAFFOLD can draw multiple batches during a round). The result is that (as described in Subsection 5.2), 1-WAY GRADIENT TRANSFER works out to be more sensitive to central variance $\sigma_c^2$ and number of steps $K$ than 2-WAY GRADIENT TRANSFER.

We will first state a number of definitions and lemmas in Subsection D.1 that are needed in proving convergence rate of 1-WAY GRADIENT TRANSFER, before proceeding to the actual proofs in Subsection D.2.

### D.1 ADDITIONAL DEFINITIONS AND LEMMAS

Note that some of the lemmas below are restatements of lemmas given in Karimireddy et al. (2020b). We opt to restate here (versus referencing the relevant lemma in Karimireddy et al. (2020b) each time) due to the volume of usage of the lemmas, to ease the burden on the reader.

We will first present the subset of definitions and lemmas which don't make any assumptions of convexity (Subsection D.1.1), followed by the subset that assume convexity (Subsection D.1.2)

### D.1.1 GENERAL DEFINITIONS AND LEMMAS

**Definition D.1** ($\beta$-Smoothness). A function $h$ is $\beta$-**smooth** if it satisfies:
$$\|\nabla h(\boldsymbol{x}) - \nabla h(\boldsymbol{y})\| \leq \beta \|\boldsymbol{x} - \boldsymbol{y}\|, \text{ for any } \boldsymbol{x}, \boldsymbol{y}$$
This implies the following quadratic upper bound on $h$:
$$\langle \nabla h(\boldsymbol{x}), \boldsymbol{y} - \boldsymbol{x} \rangle \geq - \left( h(\boldsymbol{x}) - h(\boldsymbol{y}) + \frac{\beta}{2} \|\boldsymbol{x} - \boldsymbol{y}\|^2 \right), \text{ for any } \boldsymbol{x}, \boldsymbol{y}$$

**Lemma D.2** (Relaxed triangle inequality). *Let $\{v_1, \dots, v_\tau\}$ be $\tau$ vectors in $\mathbb{R}^d$. Then for any $a > 0$:*
$$\|v_i + v_j\|^2 \leq (1 + a) \|v_i\|^2 + \left(1 + \frac{1}{a}\right) \|v_j\|^2$$
*Also:*
$$\left\| \sum_{i=1}^{\tau} v_i \right\|^2 \leq \tau \sum_{i=1}^{\tau} \|v_i\|^2$$

*Proof.* The first statement for any $a > 0$ follows from the identity:
$$\|v_i + v_j\|^2 = (1 + a) \|v_i\|^2 + \left(1 + \frac{1}{a}\right) \|v_j\|^2 - \left\| \sqrt{a} v_i + \frac{1}{\sqrt{a}} v_j \right\|^2$$
The second statement follows from the convexity of $v \to \|v\|^2$ and Jensen's inequality:
$$\left\| \frac{1}{\tau} \sum_{i=1}^{\tau} v_i \right\|^2 \leq \frac{1}{\tau} \sum_{i=1}^{\tau} \|v_i\|^2$$
$\square$

**Lemma D.3** (Separating mean and variance). *Let $\{\Xi_1, \dots, \Xi_\tau\}$ be $\tau$ random variables in $\mathbb{R}^d$ which are not necessarily independent. First suppose that their mean is $\mathbb{E}[\Xi_i] = \xi_i$ and variance is bounded as $\mathbb{E}[\|\Xi_i - \xi_i\|^2] \leq \sigma^2$. Then:*
$$\mathbb{E}\left[ \left\| \sum_{i=1}^{\tau} \Xi_i \right\|^2 \right] \leq \left\| \sum_{i=1}^{\tau} \xi_i \right\|^2 + \tau^2 \sigma^2$$

*Now instead suppose that their* conditional *mean is* $\mathbb{E}[\Xi_i | \Xi_{i-1}, \ldots, \Xi_1] = \xi_i$, *i.e. the variables* $\{\Xi_i - \xi_i\}$ *form a martingale difference sequence, and the variance is bounded same as above. Then we can show the tighter bound:*

$$\mathbb{E}\left[\left\|\sum_{i=1}^{\tau} \Xi_i\right\|^2\right] \leq 2\mathbb{E}\left[\left\|\sum_{i=1}^{\tau} \xi_i\right\|^2\right] + 2\tau\sigma^2$$

*Proof.* For any random variable $X$, $\mathbb{E}[X^2] = (\mathbb{E}[X - \mathbb{E}[X]])^2 + (\mathbb{E}[X])^2$ implying:

$$\mathbb{E}\left[\left\|\sum_{i=1}^{\tau} \Xi_i\right\|^2\right] = \left\|\sum_{i=1}^{\tau} \xi_i\right\|^2 + \mathbb{E}\left[\left\|\sum_{i=1}^{\tau} (\Xi_i - \xi_i)\right\|^2\right]$$

Expanding the last term of the above expression using relaxed triangle inequality (Lemma D.2) proves the first claim:

$$\mathbb{E}\left[\left\|\sum_{i=1}^{\tau} (\Xi_i - \xi_i)\right\|^2\right] \leq \tau \sum_{i=1}^{\tau} \mathbb{E}\left[\|\Xi_i - \xi_i\|^2\right] \leq \tau^2 \sigma^2$$

For the second statement, $\xi_i$ is not deterministic and depends on $\Xi_{i-1}, \ldots, \Xi_1$. Hence we have to resort to the cruder relaxed triangle inequality to claim:

$$\mathbb{E}\left[\left\|\sum_{i=1}^{\tau} \Xi_i\right\|^2\right] \leq 2\mathbb{E}\left[\left\|\sum_{i=1}^{\tau} \xi_i\right\|^2\right] + 2\mathbb{E}\left[\left\|\sum_{i=1}^{\tau} (\Xi_i - \xi_i)\right\|^2\right]$$

Then we use the tighter expansion of the second term:

$$\mathbb{E}\left[\left\|\sum_{i=1}^{\tau} (\Xi_i - \xi_i)\right\|^2\right] = \sum_{i,j} \mathbb{E}\left[\langle \Xi_i - \xi_i, \Xi_j - \xi_j \rangle\right] = \sum_i \mathbb{E}\left[\|\Xi_i - \xi_i\|^2\right] \leq \tau\sigma^2$$

The cross terms in the above expression have zero mean since $\{\Xi_i - \xi_i\}$ form a martingale difference sequence. $\square$

### D.1.2 DEFINITIONS AND LEMMAS ASSUMING CONVEXITY

**Definition D.4** ($\mu$-Convexity). A function $h$ is $\mu$-**convex** for $\mu \geq 0$ if it satisfies:

$$\langle \nabla h(\boldsymbol{x}), \boldsymbol{y} - \boldsymbol{x} \rangle \leq - \left( h(\boldsymbol{x}) - h(\boldsymbol{y}) + \frac{\mu}{2} \|\boldsymbol{x} - \boldsymbol{y}\|^2 \right), \text{ for any } \boldsymbol{x}, \boldsymbol{y}$$

When $\mu > 0$, we have strong convexity, a quadratic lower bound on $h$.

**Proposition D.5** (Convexity and smoothness). *If client losses $f_i$ and centralized loss $f_c$ are each $\beta$-smooth (Definition D.1), and $\boldsymbol{x}^*$ is an optimum of the overall loss $f$ (as defined in Equation 1), then the following holds true:*

$$\frac{1}{2\beta} \left( \frac{1}{N} \sum_{i=1}^{N} \|\nabla f_i(\boldsymbol{x}) - \nabla f_i(\boldsymbol{x}^*)\|^2 + \|\nabla f_c(\boldsymbol{x}) - \nabla f_c(\boldsymbol{x}^*)\|^2 \right) \leq f(\boldsymbol{x}) - f(\boldsymbol{x}^*)$$

*Proof.* Define the functions $\tilde{f}_i(\boldsymbol{x}) := f_i(\boldsymbol{x}) - \langle \nabla f_i(\boldsymbol{x}^*), \boldsymbol{x} \rangle$, for all clients $i$, and the function $\tilde{f}_c(\boldsymbol{x}) := f_c(\boldsymbol{x}) - \langle \nabla f_c(\boldsymbol{x}^*), \boldsymbol{x} \rangle$. Since $f_i$ and $f_c$ are convex and $\beta$-smooth, so are $\tilde{f}_i$ and $\tilde{f}_c$, and furthermore their gradients vanish at $\boldsymbol{x}^*$; hence, $\boldsymbol{x}^*$ is a common minimizer for $\tilde{f}_i$, $\tilde{f}_c$ and $f$. Using the $\beta$-smoothness of $\tilde{f}_i$ and $f_c$, we have

$$\frac{1}{2\beta} \|\nabla \tilde{f}_i(\boldsymbol{x})\|^2 \leq \tilde{f}_i(\boldsymbol{x}) - \tilde{f}_i(\boldsymbol{x}) \quad \text{and} \quad \frac{1}{2\beta} \|\nabla \tilde{f}_c(\boldsymbol{x})\|^2 \leq \tilde{f}_c(\boldsymbol{x}) - \tilde{f}_c(\boldsymbol{x}).$$

Note that $\frac{1}{N} \sum_{i=1}^{N} \tilde{f}_i + \tilde{f}_c = f$ since $\frac{1}{N} \sum_{i=1}^{N} \nabla f_i(\boldsymbol{x}^*) + \nabla f_c(\boldsymbol{x}^*) = \nabla f(\boldsymbol{x}^*) = 0$. The claimed bound then follows from the above two facts. $\square$

**Proposition D.6** (Convex bound on gradient of overall loss). *If client losses $f_i$ and centralized loss $f_c$ are each $\mu$-convex (Definition D.4) and $\beta$-smooth (Definition D.1), and $\boldsymbol{x}^*$ is an optimum of the overall loss $f$ (as defined in Equation 1), then the expected norm of the gradient of overall loss is bounded as:*

$$\mathbb{E} \left\| \nabla f(\boldsymbol{x}) \right\|^2 \leq 4\beta \mathbb{E} \left[ f(\boldsymbol{x}) - f(\boldsymbol{x}^*) \right]$$

*Proof.*

$$\mathbb{E} \left\| \nabla f(\boldsymbol{x}) \right\|^2 = \mathbb{E} \left\| \nabla f(\boldsymbol{x}) - \nabla f(\boldsymbol{x}^*) \right\|^2$$

$$= \mathbb{E} \left\| \frac{1}{N} \sum_{i=1}^{N} \left( \nabla f_i(\boldsymbol{x}) - \nabla f_i(\boldsymbol{x}^*) \right) + \left( \nabla f_c(\boldsymbol{x}) - \nabla f_c(\boldsymbol{x}^*) \right) \right\|^2$$

Applying the relaxed triangle inequality (Lemma D.2) twice:

$$\mathbb{E} \left\| \nabla f(\boldsymbol{x}) \right\|^2 \leq 2\mathbb{E} \left\| \frac{1}{N} \sum_{i=1}^{N} \left( \nabla f_i(\boldsymbol{x}) - \nabla f_i(\boldsymbol{x}^*) \right) \right\|^2 + 2\mathbb{E} \left\| \nabla f_c(\boldsymbol{x}) - \nabla f_c(\boldsymbol{x}^*) \right\|^2$$

$$\leq 2\mathbb{E} \frac{1}{N} \sum_{i=1}^{N} \left\| \left( \nabla f_i(\boldsymbol{x}) - \nabla f_i(\boldsymbol{x}^*) \right) \right\|^2 + 2\mathbb{E} \left\| \nabla f_c(\boldsymbol{x}) - \nabla f_c(\boldsymbol{x}^*) \right\|^2$$

$$\leq 2\mathbb{E} \left[ \frac{1}{N} \sum_{i=1}^{N} \left\| \left( \nabla f_i(\boldsymbol{x}) - \nabla f_i(\boldsymbol{x}^*) \right) \right\|^2 + \left\| \nabla f_c(\boldsymbol{x}) - \nabla f_c(\boldsymbol{x}^*) \right\|^2 \right]$$

Applying Proposition D.5:

$$\mathbb{E} \left\| \nabla f(\boldsymbol{x}) \right\|^2 \leq 2\mathbb{E} \left[ 2\beta \left( f(\boldsymbol{x}) - f(\boldsymbol{x}^*) \right) \right]$$
$$\leq 4\beta \mathbb{E} \left[ f(\boldsymbol{x}) - f(\boldsymbol{x}^*) \right]$$

$\square$

**Lemma D.7** (Perturbed strong convexity). *The following holds for any $\beta$-smooth and $\mu$-strongly-convex function $h$, and any $\boldsymbol{x}, \boldsymbol{y}, \boldsymbol{z}$ in the domain of $h$:*

$$\langle \nabla h(\boldsymbol{x}), \boldsymbol{z} - \boldsymbol{y} \rangle \geq h(\boldsymbol{z}) - h(\boldsymbol{y}) + \frac{\mu}{4} \left\| \boldsymbol{y} - \boldsymbol{z} \right\|^2 - \beta \left\| \boldsymbol{z} - \boldsymbol{x} \right\|^2$$

*Proof.* Given any $\boldsymbol{x}, \boldsymbol{y}$, and $\boldsymbol{z}$, we get the following two inequalities using smoothness (Definition D.1) and strong convexity (Definition D.4) of $h$:

$$\langle \nabla h(\boldsymbol{x}), \boldsymbol{z} - \boldsymbol{x} \rangle \geq h(\boldsymbol{z}) - h(\boldsymbol{x}) - \frac{\beta}{2} \left\| \boldsymbol{z} - \boldsymbol{x} \right\|^2$$
$$\geq h(\boldsymbol{x}) - h(\boldsymbol{y}) + \frac{\mu}{2} \left\| \boldsymbol{y} - \boldsymbol{x} \right\|^2$$

Further, applying the relaxed triangle inequality (Lemma D.2) gives:

$$\frac{\mu}{2} \left\| \boldsymbol{y} - \boldsymbol{x} \right\|^2 \geq \frac{\mu}{4} \left\| \boldsymbol{y} - \boldsymbol{z} \right\|^2 - \frac{\mu}{2} \left\| \boldsymbol{x} - \boldsymbol{z} \right\|^2$$

Combining all the inequalities together we have:

$$\langle \nabla h(\boldsymbol{x}), \boldsymbol{z} - \boldsymbol{y} \rangle \geq h(\boldsymbol{z}) - h(\boldsymbol{y}) + \frac{\mu}{4} \left\| \boldsymbol{y} - \boldsymbol{z} \right\|^2 - \frac{\beta + \mu}{2} \left\| \boldsymbol{z} - \boldsymbol{x} \right\|^2$$

The lemma follows since $\beta \geq \mu$. $\square$

**Lemma D.8** (Contractive mapping). *For any $\beta$-smooth and $\mu$-strongly convex function $h$, values $\boldsymbol{x}$ and $\boldsymbol{y}$ in the domain of $h$, and step-size (learning rate) $\eta \leq \frac{1}{\beta}$, the following holds true:*

$$\left\| \boldsymbol{x} - \eta \nabla h(\boldsymbol{x}) - \boldsymbol{y} + \eta \nabla h(\boldsymbol{y}) \right\|^2 \leq (1 - \mu\eta) \left\| \boldsymbol{x} - \boldsymbol{y} \right\|^2$$

*Proof.* Expanding terms, and applying smoothness (Definition D.1):

$$\|\boldsymbol{x} - \eta\nabla h(\boldsymbol{x}) - \boldsymbol{y} + \eta\nabla h(\boldsymbol{y})\|^2 = \|\boldsymbol{x} - \boldsymbol{y}\|^2 + \eta^2 \|\nabla h(\boldsymbol{x}) - \nabla h(\boldsymbol{y})\|^2$$
$$- 2\eta\langle\nabla h(\boldsymbol{x}) - \nabla h(\boldsymbol{y}), \boldsymbol{x} - \boldsymbol{y}\rangle$$
$$\leq \|\boldsymbol{x} - \boldsymbol{y}\|^2 + \left(\eta^2\beta - 2\eta\right)\langle\nabla h(\boldsymbol{x}) - \nabla h(\boldsymbol{y}), \boldsymbol{x} - \boldsymbol{y}\rangle$$

If step-size is such that $\eta \leq \frac{1}{\beta}$, then:

$$\left(\eta^2\beta - 2\eta\right)\langle\nabla h(\boldsymbol{x}) - \nabla h(\boldsymbol{y}), \boldsymbol{x} - \boldsymbol{y}\rangle \leq -\eta\langle\nabla h(\boldsymbol{x}) - \nabla h(\boldsymbol{y}), \boldsymbol{x} - \boldsymbol{y}\rangle$$

Finally, for $\mu$-strong convexity (Definition D.4) of $h$ we have:

$$-\eta\langle\nabla h(\boldsymbol{x}) - \nabla h(\boldsymbol{y}), \boldsymbol{x} - \boldsymbol{y}\rangle \leq -\mu\eta \|\boldsymbol{x} - \boldsymbol{y}\|^2$$

$\square$

## D.2    PROOFS OF THEOREM C.2

We will now prove the rates of convergence stated in Theorem C.2 for 1-WAY GRADIENT TRANSFER. Subsection D.2.1 proves the convergence rates for strongly convex and general convex cases, and Subsection D.2.2 proves the convergence rates for the non-convex case.

Let $S$ be the cardinality of the cohort of clients $\mathcal{S}$ participating in a round of training. Let the server and client optimizers be SGD. Let the clients all take an equal number of steps $K$, and let $\tilde{\eta}$ be the 'effective step-size', equal to $K\eta_s\eta$. With 1-WAY GRADIENT TRANSFER, the server update of the global model at round $t$ can be written as:

$$\boldsymbol{x}^{(t+1)} - \boldsymbol{x}^{(t)} = -\frac{\tilde{\eta}}{KS}\sum_{i\in\mathcal{S}}\sum_{k=1}^{K}\left(g_i(\boldsymbol{x}_i^{(t,k)}) + g_c(\boldsymbol{x}^{(t)})\right)$$

$$\boldsymbol{x}^{(t+1)} - \boldsymbol{x}^{(t)} = -\tilde{\eta}g_c(\boldsymbol{x}^{(t)}) - \frac{\tilde{\eta}}{KS}\sum_{i\in\mathcal{S}}\sum_{k=1}^{K}\left(g_i(\boldsymbol{x}_i^{(t,k)})\right) \tag{11}$$

Henceforth, let $\mathbb{E}_{|t}[\cdot]$ denote expectation conditioned on $\boldsymbol{x}^{(t)}$. As in Karimireddy et al. (2020b), we'll define a client local 'drift' term in round $t$ as:

$$\mathcal{E}^{(t)} = \frac{1}{KN}\sum_{i=1}^{N}\sum_{k=1}^{K}\mathbb{E}_{|t}\left\|\boldsymbol{x}_i^{(t,k)} - \boldsymbol{x}^{(t)}\right\|^2 \tag{12}$$

**Lemma D.9** (Bound on variance of server update). *The variance of the server update is bounded as:*

$$\mathbb{E}_{|t}\left[\left\|\boldsymbol{x}^{(t+1)} - \boldsymbol{x}^{(t)}\right\|^2\right] \leq 4\tilde{\eta}^2\beta^2\mathcal{E}^{(t)} + 2\tilde{\eta}^2\left(\frac{2}{KS}\sigma^2 + \sigma_c^2\right) + 2\tilde{\eta}^2\mathbb{E}_{|t}\left[\left\|\nabla f(\boldsymbol{x}^{(t)})\right\|^2\right]$$

*Proof.* Let $\mathcal{S}$ denote the set of clients sampled in round $t$. For brevity, we will use $\Delta\boldsymbol{x}$ to refer to $\boldsymbol{x}^{(t+1)} - \boldsymbol{x}^{(t)}$.

$$\mathbb{E}_{|t}\|\Delta\boldsymbol{x}\|^2 = \mathbb{E}_{|t}\left[\left\|\boldsymbol{x}^{(t+1)} - \boldsymbol{x}^{(t)}\right\|^2\right]$$

$$= \mathbb{E}_{|t}\left[\left\|\frac{\tilde{\eta}}{KS}\sum_{i\in\mathcal{S}}\sum_{k=1}^{K}\left(g_i(\boldsymbol{x}_i^{(t,k)}) + g_c(\boldsymbol{x}^{(t)})\right)\right\|^2\right]$$

$$\leq \mathbb{E}_{|t}\left[\left\|\frac{\tilde{\eta}}{KS}\sum_{i\in\mathcal{S}}\sum_{k=1}^{K}\left(g_i(\boldsymbol{x}_i^{(t,k)}) - \nabla f_f(\boldsymbol{x}^{(t)})\right) + \left(\nabla f_f(\boldsymbol{x}^{(t)}) + g_c(\boldsymbol{x}^{(t)})\right)\right\|^2\right]$$

We separate terms by applying the relaxed triangle inequality (Lemma D.2):

$$\mathbb{E}_{|t} \left\| \Delta \boldsymbol{x} \right\|^2 \leq \underbrace{2\tilde{\eta}^2 \mathbb{E}_{|t} \left[ \left\| \frac{1}{KS} \sum_{i \in \mathcal{S}} \sum_{k=1}^{K} \left( g_i(\boldsymbol{x}_i^{(t,k)}) - \nabla f_{\mathrm{f}}(\boldsymbol{x}^{(t)}) \right) \right\|^2 \right]}_{\mathcal{A}} + \underbrace{2\tilde{\eta}^2 \mathbb{E}_{|t} \left[ \left\| \nabla f_{\mathrm{f}}(\boldsymbol{x}^{(t)}) + g_{\mathrm{c}}(\boldsymbol{x}^{(t)}) \right\|^2 \right]}_{\mathcal{B}}$$

In term $\mathcal{A}$, we separate mean and variance for the client stochastic gradients $g_i$, using Lemma D.3 and Equation 4:

$$\mathcal{A} \leq 4\tilde{\eta}^2 \mathbb{E}_{|t} \left[ \left\| \frac{1}{KS} \sum_{i \in \mathcal{S}} \sum_{k=1}^{K} \left( \nabla f_{\mathrm{f}}(\boldsymbol{x}_i^{(t,k)}) - \nabla f_{\mathrm{f}}(\boldsymbol{x}^{(t)}) \right) \right\|^2 \right] + \frac{4\tilde{\eta}^2 \sigma^2}{KS}$$

We apply the relaxed triangle inequality (Lemma D.2) followed by smoothness (Definition D.1), to convert it to an expression in terms of drift $\mathcal{E}^{(t)}$:

$$\begin{aligned}
\mathcal{A} &\leq \frac{4\tilde{\eta}^2}{KN} \sum_{i=1}^{N} \sum_{k=1}^{K} \mathbb{E}_{|t} \left[ \left\| \nabla f_{\mathrm{f}}(\boldsymbol{x}_i^{(t,k)}) - \nabla f_{\mathrm{f}}(\boldsymbol{x}^{(t)}) \right\|^2 \right] + \frac{4\tilde{\eta}^2 \sigma^2}{KS} \\
&\leq \frac{4\tilde{\eta}^2 \beta^2}{KN} \sum_{i=1}^{N} \sum_{k=1}^{K} \mathbb{E}_{|t} \left[ \left\| \boldsymbol{x}_i^{(t,k)} - \boldsymbol{x}^{(t)} \right\|^2 \right] + \frac{4\tilde{\eta}^2 \sigma^2}{KS} \\
&\leq 4\tilde{\eta}^2 \beta^2 \mathcal{E}^{(t)} + \frac{4\tilde{\eta}^2 \sigma^2}{KS}
\end{aligned}$$

In term $\mathcal{B}$ we have a full gradient of the federated loss $\nabla f_{\mathrm{f}}$ and a stochastic gradient of the centralized loss $g_{\mathrm{c}}$. We use Lemma D.3 to separate the stochastic gradient into a full gradient of the centralized loss $\nabla f_{\mathrm{c}}$ and a variance term, allowing us to express in terms of full gradient of the overall loss $\nabla f$.

$$\begin{aligned}
\mathcal{B} &= 2\tilde{\eta}^2 \mathbb{E}_{|t} \left[ \left\| \nabla f_{\mathrm{f}}(\boldsymbol{x}^{(t)}) + g_{\mathrm{c}}(\boldsymbol{x}^{(t)}) \right\|^2 \right] \\
&\leq 2\tilde{\eta}^2 \mathbb{E}_{|t} \left[ \left\| \nabla f_{\mathrm{f}}(\boldsymbol{x}^{(t)}) + \nabla f_{\mathrm{c}}(\boldsymbol{x}^{(t)}) \right\|^2 \right] + 2\tilde{\eta}^2 \sigma_{\mathrm{c}}^2 \\
&\leq 2\tilde{\eta}^2 \mathbb{E}_{|t} \left[ \left\| \nabla f(\boldsymbol{x}^{(t)}) \right\|^2 \right] + 2\tilde{\eta}^2 \sigma_{\mathrm{c}}^2
\end{aligned}$$

Combining $\mathcal{A}$ and $\mathcal{B}$ back together:

$$\mathbb{E}_{|t} \left\| \Delta \boldsymbol{x} \right\|^2 \leq 4\tilde{\eta}^2 \beta^2 \mathcal{E}^{(t)} + 2\tilde{\eta}^2 \left( \frac{2}{KS} \sigma^2 + \sigma_{\mathrm{c}}^2 \right) + 2\tilde{\eta}^2 \mathbb{E}_{|t} \left[ \left\| \nabla f(\boldsymbol{x}^{(t)}) \right\|^2 \right]$$

$\square$

### D.2.1 CONVEX CASES

We will state two lemmas, one (Lemma D.10) related to the progress in round $t$ towards reaching $\boldsymbol{x}^*$, and the other (Lemma D.11) bounding the federated clients 'drift' in round $t$, $\mathcal{E}^{(t)}$. We then combine the two lemmas together to give the proofs of convergence rate for the strongly convex ($\mu > 0$) and general convex ($\mu = 0$) cases.

**Lemma D.10** (One round progress). *Suppose our functions satisfy bounded variance $\sigma^2$, $\mu$-convexity (Definition D.4), and $\beta$-smoothness (Definition D.1). If $\tilde{\eta} < \frac{1}{8\beta}$, the updates of* 1-WAY GRADIENT TRANSFER *satisfy:*

$$\mathbb{E}_{|t}\left[\left\|\boldsymbol{x}^{(t+1)} - \boldsymbol{x}^*\right\|^2\right] \le \left(1 - \frac{3\mu\tilde{\eta}}{2}\right)\left\|\boldsymbol{x}^{(t)} - \boldsymbol{x}^*\right\|^2 - \tilde{\eta}\left(f(\boldsymbol{x}^{(t)}) - f(\boldsymbol{x}^*)\right) + \frac{5}{16}\mathcal{E}^{(t)}$$
$$+ 2\tilde{\eta}^2\left(\frac{2}{KS}\sigma^2 + \sigma_c^2\right)$$

*Proof.* The expected server update, with $N$ total clients in the federated population, is:

$$\mathbb{E}\left[\boldsymbol{x}^{(t+1)} - \boldsymbol{x}^{(t)}\right] = -\tilde{\eta}\mathbb{E}\left[g_{\mathrm{c}}(\boldsymbol{x}^{(t)})\right] - \frac{\tilde{\eta}}{KN}\sum_{i=1}^{N}\sum_{k=1}^{K}\mathbb{E}\left[g_i(\boldsymbol{x}_i^{(t,k)})\right]$$

The distance from optimal $\boldsymbol{x}^*$ in parameter space at round $t$ is $\left\|\boldsymbol{x}^{(t)} - \boldsymbol{x}^*\right\|^2$. The expected distance from optimal at round $t+1$, conditioned on $\boldsymbol{x}^{(t)}$ and earlier rounds, is:

$$\mathbb{E}_{|t}\left[\left\|\boldsymbol{x}^{(t+1)} - \boldsymbol{x}^*\right\|^2\right] = \mathbb{E}_{|t}\left[\left\|\boldsymbol{x}^{(t+1)} - \boldsymbol{x}^{(t)} + \boldsymbol{x}^{(t)} - \boldsymbol{x}^*\right\|^2\right]$$
$$= \left\|\boldsymbol{x}^{(t)} - \boldsymbol{x}^*\right\|^2 + \underbrace{2\left\langle\mathbb{E}_{|t}\left[\boldsymbol{x}^{(t+1)} - \boldsymbol{x}^{(t)}\right], \boldsymbol{x}^{(t)} - \boldsymbol{x}^*\right\rangle}_{\mathcal{C}} + \underbrace{\mathbb{E}_{|t}\left[\left\|\boldsymbol{x}^{(t+1)} - \boldsymbol{x}^{(t)}\right\|^2\right]}_{\mathcal{D}}$$

For clarity, we now focus on individual terms, beginning with $\mathcal{C}$:

$$\mathcal{C} = 2\left\langle\mathbb{E}_{|t}\left[\boldsymbol{x}^{(t+1)} - \boldsymbol{x}^{(t)}\right], \boldsymbol{x}^{(t)} - \boldsymbol{x}^*\right\rangle$$
$$= 2\left\langle\left(-\tilde{\eta}\mathbb{E}\left[g_{\mathrm{c}}(\boldsymbol{x}^{(t)})\right] - \frac{\tilde{\eta}}{KN}\sum_{i=1}^{N}\sum_{k=1}^{K}\mathbb{E}\left[g_i(\boldsymbol{x}_i^{(t,k)})\right]\right), \boldsymbol{x}^{(t)} - \boldsymbol{x}^*\right\rangle$$
$$= \underbrace{2\tilde{\eta}\left\langle\nabla f_{\mathrm{c}}(\boldsymbol{x}^{(t)}), \boldsymbol{x}^* - \boldsymbol{x}^{(t)}\right\rangle}_{\mathcal{C}1} + \underbrace{\frac{2\tilde{\eta}}{KN}\left\langle\sum_{i=1}^{N}\sum_{k=1}^{K}\nabla f_{\mathrm{f}}(\boldsymbol{x}_i^{(t,k)}), \boldsymbol{x}^* - \boldsymbol{x}^{(t)}\right\rangle}_{\mathcal{C}2}$$

We can use convexity (Definition D.4) to bound $\mathcal{C}1$, with $\boldsymbol{x} = \boldsymbol{x}^{(t)}$, and $y = \boldsymbol{x}^*$:

$$\mathcal{C}1 \le -2\tilde{\eta}\left(f_{\mathrm{c}}(\boldsymbol{x}^{(t)}) - f_{\mathrm{c}}(\boldsymbol{x}^*) + \frac{\mu}{2}\left\|\boldsymbol{x}^{(t)} - \boldsymbol{x}^*\right\|^2\right)$$

We apply perturbed convexity (Lemma D.7) to bound $\mathcal{C}2$, with $\boldsymbol{x} = \boldsymbol{x}_i^{(t,k)}$, $\boldsymbol{y} = \boldsymbol{x}^*$, and $\boldsymbol{z} = \boldsymbol{x}^{(t)}$:

$$\mathcal{C}2 \le \frac{2\tilde{\eta}}{KN}\sum_{i=1}^{N}\sum_{k=1}^{K}\left(f_{\mathrm{f}}(\boldsymbol{x}^*) - f_{\mathrm{f}}(\boldsymbol{x}^{(t)}) + \beta\left\|\boldsymbol{x}_i^{(t,k)} - \boldsymbol{x}^{(t)}\right\|^2 - \frac{\mu}{4}\left\|\boldsymbol{x}^{(t)} - \boldsymbol{x}^*\right\|^2\right)$$
$$\le -2\tilde{\eta}\left(f_{\mathrm{f}}(\boldsymbol{x}^{(t)}) - f_{\mathrm{f}}(\boldsymbol{x}^*) + \frac{\mu}{4}\left\|\boldsymbol{x}^{(t)} - \boldsymbol{x}^*\right\|^2\right) + \frac{2\beta\tilde{\eta}}{KN}\sum_{i=1}^{N}\sum_{k=1}^{K}\left\|\boldsymbol{x}_i^{(t,k)} - \boldsymbol{x}^{(t)}\right\|^2$$
$$\le -2\tilde{\eta}\left(f_{\mathrm{f}}(\boldsymbol{x}^{(t)}) - f_{\mathrm{f}}(\boldsymbol{x}^*) + \frac{\mu}{4}\left\|\boldsymbol{x}^{(t)} - \boldsymbol{x}^*\right\|^2\right) + 2\beta\tilde{\eta}\mathcal{E}^{(t)}$$

Combining $\mathcal{C}1$ and $\mathcal{C}2$ back together:

$$\mathcal{C} \leq -2\tilde{\eta}\left(f(\boldsymbol{x}^{(t)}) - f(\boldsymbol{x}^*) + \frac{3\mu}{4}\left\|\boldsymbol{x}^{(t)} - \boldsymbol{x}^*\right\|^2\right) + 2\beta\tilde{\eta}\mathcal{E}^{(t)}$$

Now we turn to term $\mathcal{D}$, which is the variance of the server update (from Lemma D.9):

$$\mathcal{D} = \mathbb{E}_{|t}\left[\left\|\boldsymbol{x}^{(t+1)} - \boldsymbol{x}^{(t)}\right\|^2\right] \leq 4\tilde{\eta}^2\beta^2\mathcal{E}^{(t)} + 2\tilde{\eta}^2\left(\frac{2}{KS}\sigma^2 + \sigma_c^2\right) + 2\tilde{\eta}^2\mathbb{E}_{|t}\left[\left\|\nabla f(\boldsymbol{x}^{(t)})\right\|^2\right]$$

We can leverage Proposition D.6 to replace the norm squared of the gradient of the overall loss:

$$\mathcal{D} = \mathbb{E}_{|t}\left[\left\|\boldsymbol{x}^{(t+1)} - \boldsymbol{x}^{(t)}\right\|^2\right] \leq 4\tilde{\eta}^2\beta^2\mathcal{E}^{(t)} + 2\tilde{\eta}^2\left(\frac{2}{KS}\sigma^2 + \sigma_c^2\right) + 8\tilde{\eta}^2\beta\mathbb{E}_{|t}\left[f(\boldsymbol{x}^{(t)}) - f(\boldsymbol{x}^*)\right]$$

Returning to our equation for the expected distance from optimal $\boldsymbol{x}^*$ in parameter space, and making use of the bounds we established for $\mathcal{C}$ and $\mathcal{D}$:

$$\mathbb{E}_{|t}\left[\left\|\boldsymbol{x}^{(t+1)} - \boldsymbol{x}^*\right\|^2\right] = \left\|\boldsymbol{x}^{(t)} - \boldsymbol{x}^*\right\|^2 + \underbrace{2\left\langle\mathbb{E}_{|t}\left[\boldsymbol{x}^{(t+1)} - \boldsymbol{x}^{(t)}\right], \boldsymbol{x}^{(t)} - \boldsymbol{x}^*\right\rangle}_{\mathcal{C}} + \underbrace{\mathbb{E}_{|t}\left[\left\|\boldsymbol{x}^{(t+1)} - \boldsymbol{x}^{(t)}\right\|^2\right]}_{\mathcal{D}}$$

$$\leq \left\|\boldsymbol{x}^{(t)} - \boldsymbol{x}^*\right\|^2 - 2\tilde{\eta}\left(f(\boldsymbol{x}^{(t)}) - f(\boldsymbol{x}^*) + \frac{3\mu}{4}\left\|\boldsymbol{x}^{(t)} - \boldsymbol{x}^*\right\|^2\right)$$

$$+ 2\beta\tilde{\eta}\mathcal{E}^{(t)} + 4\tilde{\eta}^2\beta^2\mathcal{E}^{(t)} + 2\tilde{\eta}^2\left(\frac{2}{KS}\sigma^2 + \sigma_c^2\right)$$

$$+ 8\tilde{\eta}^2\beta\mathbb{E}_{|t}\left[f(\boldsymbol{x}^{(t)}) - f(\boldsymbol{x}^*)\right]$$

$$\leq \left(1 - \frac{3\mu\tilde{\eta}}{2}\right)\left\|\boldsymbol{x}^{(t)} - \boldsymbol{x}^*\right\|^2$$

$$+ \left(8\tilde{\eta}^2\beta - 2\tilde{\eta}\right)\left(f(\boldsymbol{x}^{(t)}) - f(\boldsymbol{x}^*)\right)$$

$$+ 2\tilde{\eta}\beta\left(1 + 2\tilde{\eta}\beta\right)\mathcal{E}^{(t)} + 2\tilde{\eta}^2\left(\frac{2}{KS}\sigma^2 + \sigma_c^2\right)$$

Assuming that $\tilde{\eta} \leq \frac{1}{8\beta}$:

$$\mathbb{E}_{|t}\left[\left\|\boldsymbol{x}^{(t+1)} - \boldsymbol{x}^*\right\|^2\right] \leq \left(1 - \frac{3\mu\tilde{\eta}}{2}\right)\left\|\boldsymbol{x}^{(t)} - \boldsymbol{x}^*\right\|^2 - \tilde{\eta}\left(f(\boldsymbol{x}^{(t)}) - f(\boldsymbol{x}^*)\right) + \frac{5}{16}\mathcal{E}^{(t)}$$

$$+ 2\tilde{\eta}^2\left(\frac{2}{KS}\sigma^2 + \sigma_c^2\right)$$

$$\square$$

**Lemma D.11** (Bounded drift). *Suppose our functions satisfy bounded variance, $\mu$-convexity (Definition D.4), and $\beta$-smoothness (Definition D.1). Then the drift is bounded as:*

$$\mathcal{E}^{(t)} \leq 12K^2\eta^2\beta\mathbb{E}\left[f(\boldsymbol{x}^{(t)}) - f(\boldsymbol{x}^*)\right] + 3K^2\eta^2\left(\frac{1}{K}\sigma^2 + \sigma_c^2\right)$$

*Proof.* We begin with the summand of the drift term, looking at the drift of a particular client $i$ at local step $k$. Expanding this summand out:

$$\mathbb{E}_{|t}\left\|\boldsymbol{x}_i^{(t,k)} - \boldsymbol{x}^{(t)}\right\|^2 = \mathbb{E}_{|t}\left\|\boldsymbol{x}_i^{(t,k-1)} - \eta\left(g_i(\boldsymbol{x}_i^{(t,k-1)}) + g_c(\boldsymbol{x}^{(t)})\right) - \boldsymbol{x}^{(t)}\right\|^2$$

$$= \mathbb{E}_{|t}\left\|\boldsymbol{x}_i^{(t,k-1)} - \boldsymbol{x}^{(t)} - \eta g_i(\boldsymbol{x}_i^{(t,k-1)}) - \eta g_c(\boldsymbol{x}^{(t)})\right\|^2.$$

Separating mean and variance of the client gradient, then using the relaxed triangle inequality (Lemma D.2) to further separate out terms:

$$\mathbb{E}_{|t}\left\|\boldsymbol{x}_i^{(t,k)} - \boldsymbol{x}^{(t)}\right\|^2 \le \mathbb{E}_{|t}\left\|\boldsymbol{x}_i^{(t,k-1)} - \boldsymbol{x}^{(t)} - \eta\nabla f_f(\boldsymbol{x}_i^{(t,k-1)}) - \eta g_c(\boldsymbol{x}^{(t)})\right\|^2 + \eta^2\sigma^2$$

$$\le \left(1 + \frac{1}{a}\right)\underbrace{\mathbb{E}_{|t}\left\|\boldsymbol{x}_i^{(t,k-1)} - \boldsymbol{x}^{(t)} - \eta\left(\nabla f_f(\boldsymbol{x}_i^{(t,k-1)}) - \nabla f_f(\boldsymbol{x}^{(t)})\right)\right\|^2}_{\mathcal{F}}$$

$$+ (1+a)\eta^2\left\|\nabla f_f(\boldsymbol{x}^{(t)}) + g_c(\boldsymbol{x}^{(t)})\right\|^2 + \eta^2\sigma^2.$$

Term $\mathcal{F}$ is bounded via the contractive mapping lemma (Lemma D.8), provided that $\eta \le \frac{1}{\beta}$:

$$\mathcal{F} \le (1 - \mu\eta)\mathbb{E}_{|t}\left\|\boldsymbol{x}_i^{(t,k-1)} - \boldsymbol{x}^{(t)}\right\|^2 \le \mathbb{E}_{|t}\left\|\boldsymbol{x}_i^{(t,k-1)} - \boldsymbol{x}^{(t)}\right\|^2.$$

Putting back into the bound on drift on client $i$ at local step $k$, and letting $a = K$:

$$\mathbb{E}_{|t}\left\|\boldsymbol{x}_i^{(t,k)} - \boldsymbol{x}^{(t)}\right\|^2 \le \frac{K+1}{K}\mathbb{E}_{|t}\left\|\boldsymbol{x}_i^{(t,k-1)} - \boldsymbol{x}^{(t)}\right\|^2 + 2K\eta^2\left\|\nabla f_f(\boldsymbol{x}^{(t)}) + g_c(\boldsymbol{x}^{(t)})\right\|^2 + \eta^2\sigma^2.$$

Unrolling the recursion:

$$\mathbb{E}_{|t}\left\|\boldsymbol{x}_i^{(t,k)} - \boldsymbol{x}^{(t)}\right\|^2 \le \left(2K\eta^2\left\|\nabla f_f(\boldsymbol{x}^{(t)}) + g_c(\boldsymbol{x}^{(t)})\right\|^2 + \eta^2\sigma^2\right)\sum_{j=0}^{k-1}\left(\frac{K+1}{K}\right)^j$$

$$\le \left(2K\eta^2\left\|\nabla f_f(\boldsymbol{x}^{(t)}) + g_c(\boldsymbol{x}^{(t)})\right\|^2 + \eta^2\sigma^2\right)(2K)$$

$$\le 4K^2\eta^2\left\|\nabla f_f(\boldsymbol{x}^{(t)}) + g_c(\boldsymbol{x}^{(t)})\right\|^2 + 2K\eta^2\sigma^2.$$

The second inequality above uses the following bound:

$$\sum_{j=0}^{k-1}\left(\frac{K+1}{K}\right)^j\left((1 + \tfrac{1}{K})^k - 1\right) = K \le (e-1)K \le 2K.$$

Now separating mean and variance of the central gradient:

$$\mathbb{E}_{|t}\left\|\boldsymbol{x}_i^{(t,k)} - \boldsymbol{x}^{(t)}\right\|^2 \le 4K^2\eta^2\left\|\nabla f_f(\boldsymbol{x}^{(t)}) + \nabla f_c(\boldsymbol{x}^{(t)})\right\|^2 + 2K^2\eta^2\sigma_c^2 + 2K\eta^2\sigma^2$$

$$\le 4K^2\eta^2\left\|\nabla f(\boldsymbol{x}^{(t)})\right\|^2 + 2K^2\eta^2\left(\frac{1}{K}\sigma^2 + \sigma_c^2\right).$$

Finally, we apply Proposition D.6:

$$\mathcal{E}^{(t)} \leq 4K^2\eta^2 \left\| \nabla f(\boldsymbol{x}^{(t)}) \right\|^2 + 2K^2\eta^2 \left( \frac{1}{K}\sigma^2 + \sigma_c^2 \right)$$

$$\leq 16K^2\eta^2\beta \mathbb{E}_{|t} \left[ f(\boldsymbol{x}^{(t)}) - f(\boldsymbol{x}^*) \right] + 2K^2\eta^2 \left( \frac{1}{K}\sigma^2 + \sigma_c^2 \right).$$

Assuming that $\tilde{\eta} \leq \frac{1}{8\beta}$:

$$\mathcal{E}^{(t)} \leq 2\frac{\tilde{\eta}}{\eta_s^2} \mathbb{E}_{|t} \left[ f(\boldsymbol{x}^{(t)}) - f(\boldsymbol{x}^*) \right] + 2\frac{\tilde{\eta}^2}{\eta_s^2} \left( \frac{1}{K}\sigma^2 + \sigma_c^2 \right)$$

$\square$

**Proofs of Theorem C.2 for Convex Cases**  Adding the statements of Lemmas D.10 and D.11, and assuming that $\eta_s > \sqrt{\frac{5}{8}S}, \eta = \frac{1}{8\beta K \eta_s}$ so that $\tilde{\eta} = \frac{1}{8\beta}$, we get:

$$\begin{aligned}
\mathbb{E}_{|t} \left[ \left\| \boldsymbol{x}^{(t+1)} - \boldsymbol{x}^* \right\|^2 \right] &\leq \left( 1 - \frac{3\mu\tilde{\eta}}{2} \right) \left\| \boldsymbol{x}^{(t)} - \boldsymbol{x}^* \right\|^2 - \tilde{\eta} \left( f(\boldsymbol{x}^{(t)}) - f(\boldsymbol{x}^*) \right) \\
&\quad + \frac{5}{16} \left( 2\frac{\tilde{\eta}}{\eta_s^2} \mathbb{E}_{|t} \left[ f(\boldsymbol{x}^{(t)}) - f(\boldsymbol{x}^*) \right] + 2\frac{\tilde{\eta}^2}{\eta_s^2} \left( \frac{1}{K}\sigma^2 + \sigma_c^2 \right) \right) \\
&\quad + 2\tilde{\eta}^2 \left( \frac{2}{KS}\sigma^2 + \sigma_c^2 \right) \\
&= \left( 1 - \frac{3\mu\tilde{\eta}}{2} \right) \left\| \boldsymbol{x}^{(t)} - \boldsymbol{x}^* \right\|^2 - \tilde{\eta} \left( f(\boldsymbol{x}^{(t)}) - f(\boldsymbol{x}^*) \right) \\
&\quad + \frac{5}{8} \frac{\tilde{\eta}}{\eta_s^2} \mathbb{E}_{|t} \left[ f(\boldsymbol{x}^{(t)}) - f(\boldsymbol{x}^*) \right] \\
&\quad + \left( \frac{5}{8} \frac{\tilde{\eta}^2}{K\eta_s^2} + 4\frac{\tilde{\eta}^2}{KS} \right) \sigma^2 + \left( \frac{5}{8} \frac{\tilde{\eta}^2}{\eta_s^2} + 2\tilde{\eta}^2 \right) \sigma_c^2 \\
&\leq \left( 1 - \frac{3\mu\tilde{\eta}}{2} \right) \left\| \boldsymbol{x}^{(t)} - \boldsymbol{x}^* \right\|^2 \\
&\quad - \left( \frac{S-1}{S} \right) \tilde{\eta} \mathbb{E}_{|t} \left[ f(\boldsymbol{x}^{(t)}) - f(\boldsymbol{x}^*) \right] + \left( \frac{5\sigma^2}{KS} + 3\sigma_c^2 \right) \tilde{\eta}^2.
\end{aligned}$$

We can now remove the conditioning over $\boldsymbol{x}^{(t)}$ by taking an expectation on both sides over $\boldsymbol{x}^{(t)}$, to get a recurrence relation of the same form.

For the case of strong convexity ($\mu > 0$), we can use lemmas (e.g., Lemma 1 in Karimireddy et al. (2020b), Lemma 2 in Stich (2019)) which establish a linear convergence rate for such recursions. This results in the following bound[16] for $T \geq \frac{8\beta}{3\mu}$:

$$\mathbb{E} \left[ f(\bar{\boldsymbol{x}}^{(T)}) \right] - f(\boldsymbol{x}^*) = \tilde{\mathcal{O}} \left( \frac{\sigma^2 + KS\sigma_c^2}{\mu KST} + \mu \left\| \boldsymbol{x}^{(0)} - \boldsymbol{x}^* \right\|^2 \exp \left( \frac{-3\mu T}{16\beta} \right) \right),$$

where $\bar{\boldsymbol{x}}^{(T)}$ is a weighted average of $\boldsymbol{x}^{(1)}, \boldsymbol{x}^{(2)}, \ldots, \boldsymbol{x}^{(T+1)}$ with geometrically decreasing weights $(1 - \frac{3\mu\tilde{\eta}}{2})^{1-r}$ for $\boldsymbol{x}^{(r)}, r = 1, 2, \ldots, T+1$.

This yields an expression for the number of rounds $T$ to reach an error $\epsilon$:

$$T = \tilde{\mathcal{O}} \left( \frac{\sigma^2 + KS\sigma_c^2}{KS\mu\epsilon} + \frac{\beta}{\mu} \log \left( \frac{1}{\epsilon} \right) \right)$$

---

[16]The $\tilde{\mathcal{O}}$ notation hides dependence on logarithmic terms which can be removed by using varying step-sizes.

For the case of general convexity ($\mu = 0$), we can use lemmas (e.g., Lemma 2 in Karimireddy et al. (2020b), Lemma 4 in Stich (2019)) which establish a sublinear convergence rate for such recursions. In this case we get the following bound:

$$\mathbb{E}\left[f(\bar{\boldsymbol{x}}^{(T)})\right] - f(\boldsymbol{x}^*) \leq \left(\frac{S}{S-1}\right)\left(\frac{8\beta\left\|\boldsymbol{x}^{(0)} - \boldsymbol{x}^*\right\|^2}{T+1} + \frac{\sqrt{20\sigma^2 + 12KS\sigma_c^2}\left\|\boldsymbol{x}^{(0)} - \boldsymbol{x}^*\right\|}{\sqrt{KS(T+1)}}\right),$$

where $\bar{\boldsymbol{x}}^{(T)} = \frac{1}{T+1}\sum_{t=1}^{T+1}\boldsymbol{x}^{(t)}$.

This yields an expression for the number of rounds $T$ to reach an error $\epsilon$:

$$T = \mathcal{O}\left(\frac{(\sigma^2 + KS\sigma_c^2)D^2}{KS\epsilon^2} + \frac{\beta D^2}{\epsilon}\right).$$

In the above expression, $D^2$ is a distance in parameter space at initialization, $\left\|\boldsymbol{x}^{(0)} - \boldsymbol{x}^*\right\|^2$.

### D.2.2 Non-Convex Case

We will now prove the rate of convergence stated in Theorem C.2 for the non-convex case for 1-WAY GRADIENT TRANSFER. We will state two lemmas, one (Lemma D.12) establishing the progress made in each round, and one (Lemma D.13) bounding how much the federated clients 'drift' in a round during the course of local training. We then combine the two lemmas together give the proof of convergence rate for the non-convex case.

**Lemma D.12** (Non-convex one round progress). *The progress made in a round can be bounded as:*

$$\mathbb{E}_{|t}\left[f(\boldsymbol{x}^{(t+1)})\right] \leq f(\boldsymbol{x}^{(t)}) - \frac{4\tilde{\eta}}{9}\left\|\nabla f(\boldsymbol{x}^{(t)})\right\|^2 + \frac{\beta}{27}\mathcal{E}^{(t)} + \left(\frac{2}{KS}\sigma^2 + \sigma_c^2\right)\beta\tilde{\eta}^2$$

*Proof.* We begin by using the smoothness of $f$ to get the following bound on the expectation of $f(\boldsymbol{x}^{(t+1)})$ conditioned on $\boldsymbol{x}^{(t)}$:

$$\mathbb{E}_{|t}\left[f(\boldsymbol{x}^{(t+1)})\right] \leq \mathbb{E}_{|t}\left[f(\boldsymbol{x}^{(t)}) + \left\langle\nabla f(\boldsymbol{x}^{(t)}), \boldsymbol{x}^{(t+1)} - \boldsymbol{x}^{(t)}\right\rangle + \frac{\beta}{2}\left\|\boldsymbol{x}^{(t+1)} - \boldsymbol{x}^{(t)}\right\|^2\right]$$

$$\leq f(\boldsymbol{x}^{(t)}) + \mathbb{E}_{|t}\left\langle\nabla f(\boldsymbol{x}^{(t)}), \boldsymbol{x}^{(t+1)} - \boldsymbol{x}^{(t)}\right\rangle + \frac{\beta}{2}\mathbb{E}_{|t}\left\|\boldsymbol{x}^{(t+1)} - \boldsymbol{x}^{(t)}\right\|^2.$$

Substituting in the definition of the 1-WAY GRADIENT TRANSFER server update (Equation 11), and using Assumption 5.1 for the expectation of the client stochastic gradient:

$$\mathbb{E}_{|t}\left[f(\boldsymbol{x}^{(t+1)})\right] \leq f(\boldsymbol{x}^{(t)}) + \mathbb{E}_{|t}\left\langle\nabla f(\boldsymbol{x}^{(t)}), -\frac{\tilde{\eta}}{KS}\sum_{i\in\mathcal{S}}\sum_{k=1}^{K}\left(g_i(\boldsymbol{x}_i^{(t,k)}) + g_c(\boldsymbol{x}^{(t)})\right)\right\rangle$$

$$+ \frac{\beta}{2}\mathbb{E}_{|t}\left\|\boldsymbol{x}^{(t+1)} - \boldsymbol{x}^{(t)}\right\|^2$$

$$\leq f(\boldsymbol{x}^{(t)}) - \tilde{\eta}\left\langle\nabla f(\boldsymbol{x}^{(t)}), \frac{1}{KN}\sum_{i=1}^{N}\sum_{k=1}^{K}\left(\nabla f_f(\boldsymbol{x}_i^{(t,k)}) + \nabla f_c(\boldsymbol{x}^{(t)})\right)\right\rangle$$

$$+ \frac{\beta}{2}\mathbb{E}_{|t}\left\|\boldsymbol{x}^{(t+1)} - \boldsymbol{x}^{(t)}\right\|^2.$$

Next, we make use of the fact that $-ab = \frac{1}{2}((b-a)^2 - a^2 - b^2) \leq -\frac{1}{2}a^2 + \frac{1}{2}(b-a)^2$:

$$\mathbb{E}_{|t}\left[f(\boldsymbol{x}^{(t+1)})\right] \leq f(\boldsymbol{x}^{(t)}) - \frac{\tilde{\eta}}{2}\left\|\nabla f(\boldsymbol{x}^{(t)})\right\|^2$$

$$+ \frac{\tilde{\eta}}{2}\left\|\frac{1}{KN}\sum_{i=1}^{N}\sum_{k=1}^{K}\left(\nabla f_{\mathrm{f}}(\boldsymbol{x}_i^{(t,k)}) + \nabla f_{\mathrm{c}}(\boldsymbol{x}^{(t)})\right) - \nabla f(\boldsymbol{x}^{(t)})\right\|^2$$

$$+ \frac{\beta}{2}\mathbb{E}_{|t}\left\|\boldsymbol{x}^{(t+1)} - \boldsymbol{x}^{(t)}\right\|^2$$

$$\leq f(\boldsymbol{x}^{(t)}) - \frac{\tilde{\eta}}{2}\left\|\nabla f(\boldsymbol{x}^{(t)})\right\|^2$$

$$+ \frac{\tilde{\eta}}{2}\left\|\frac{1}{KN}\sum_{i=1}^{N}\sum_{k=1}^{K}\left(\nabla f_{\mathrm{f}}(\boldsymbol{x}_i^{(t,k)}) - \nabla f_{\mathrm{f}}(\boldsymbol{x}^{(t)})\right)\right\|^2$$

$$+ \frac{\beta}{2}\mathbb{E}_{|t}\left\|\boldsymbol{x}^{(t+1)} - \boldsymbol{x}^{(t)}\right\|^2$$

$$\leq f(\boldsymbol{x}^{(t)}) - \frac{\tilde{\eta}}{2}\left\|\nabla f(\boldsymbol{x}^{(t)})\right\|^2$$

$$+ \frac{\tilde{\eta}}{2}\frac{1}{KN}\sum_{i=1}^{N}\sum_{k=1}^{K}\mathbb{E}_{|t}\left\|\nabla f_{\mathrm{f}}(\boldsymbol{x}_i^{(t,k)}) - \nabla f_{\mathrm{f}}(\boldsymbol{x}^{(t)})\right\|^2$$

$$+ \frac{\beta}{2}\mathbb{E}_{|t}\left\|\boldsymbol{x}^{(t+1)} - \boldsymbol{x}^{(t)}\right\|^2.$$

Next, we use smoothness (Definition D.1), and the definition of client drift (Equation 12):

$$\mathbb{E}_{|t}\left[f(\boldsymbol{x}^{(t+1)})\right] \leq f(\boldsymbol{x}^{(t)}) - \frac{\tilde{\eta}}{2}\left\|\nabla f(\boldsymbol{x}^{(t)})\right\|^2$$

$$+ \frac{\tilde{\eta}\beta^2}{2}\frac{1}{KN}\sum_{i=1}^{N}\sum_{k=1}^{K}\mathbb{E}_{|t}\left\|\boldsymbol{x}_i^{(t,k)} - \boldsymbol{x}^{(t)}\right\|^2$$

$$+ \frac{\beta}{2}\mathbb{E}_{|t}\left\|\boldsymbol{x}^{(t+1)} - \boldsymbol{x}^{(t)}\right\|^2$$

$$\leq f(\boldsymbol{x}^{(t)}) - \frac{\tilde{\eta}}{2}\left\|\nabla f(\boldsymbol{x}^{(t)})\right\|^2 + \frac{\tilde{\eta}\beta^2}{2}\mathcal{E}^{(t)}$$

$$+ \frac{\beta}{2}\mathbb{E}_{|t}\left\|\boldsymbol{x}^{(t+1)} - \boldsymbol{x}^{(t)}\right\|^2.$$

The last term is the variance of the server update, for which we can substitute the bound from Lemma D.9:

$$\mathbb{E}_{|t}\left[f(\boldsymbol{x}^{(t+1)})\right] \leq f(\boldsymbol{x}^{(t)}) - \frac{\tilde{\eta}}{2}\left\|\nabla f(\boldsymbol{x}^{(t)})\right\|^2 + \frac{\tilde{\eta}\beta^2}{2}\mathcal{E}^{(t)}$$

$$+ \frac{\beta}{2}\left(4\tilde{\eta}^2\beta^2\mathcal{E}^{(t)} + 2\tilde{\eta}^2\left(\frac{2}{KS}\sigma^2 + \sigma_{\mathrm{c}}^2\right) + 2\tilde{\eta}^2\mathbb{E}_{|t}\left\|\nabla f(\boldsymbol{x}^{(t)})\right\|^2\right)$$

$$\leq f(\boldsymbol{x}^{(t)}) - \left(\frac{\tilde{\eta}}{2} - \beta\tilde{\eta}^2\right)\left\|\nabla f(\boldsymbol{x}^{(t)})\right\|^2 + \left(\frac{\tilde{\eta}\beta^2}{2} + 2\tilde{\eta}^2\beta^3\right)\mathcal{E}^{(t)}$$

$$+ \tilde{\eta}^2\beta\left(\frac{2}{KS}\sigma^2 + \sigma_{\mathrm{c}}^2\right).$$

Assuming a bound on effective step-size $\tilde{\eta} \leq \frac{1}{18\beta}$:

$$\mathbb{E}_{|t}\left[f(\boldsymbol{x}^{(t+1)})\right] \leq f(\boldsymbol{x}^{(t)}) - \frac{4\tilde{\eta}}{9}\left\|\nabla f(\boldsymbol{x}^{(t)})\right\|^2 + \frac{\beta}{27}\mathcal{E}^{(t)} + \left(\frac{2}{KS}\sigma^2 + \sigma_{\mathrm{c}}^2\right)\beta\tilde{\eta}^2.$$

$\square$

**Lemma D.13** (Non-convex bounded drift). *Suppose our functions satisfy bounded variance and $\beta$-smoothness (Definition D.1). Then the drift is bounded as:*

$$\mathcal{E}^{(t)} \leq \frac{4\tilde{\eta}}{9\beta\eta_s^2} \mathbb{E} \left\| \nabla f(\boldsymbol{x}^{(t)}) \right\|^2 + \frac{2\tilde{\eta}^2}{\eta_s^2} \left( \frac{1}{K}\sigma^2 + 4\sigma_c^2 \right).$$

*Proof.* We begin with the summand of the drift term, looking at the drift of a particular client $i$ at local step $k$. Expanding this summand out:

$$\mathbb{E} \left\| \boldsymbol{x}_i^{(t,k)} - \boldsymbol{x}^{(t)} \right\|^2 = \mathbb{E} \left\| \boldsymbol{x}_i^{(t,k-1)} - \eta \left( g_i(\boldsymbol{x}_i^{(t,k-1)}) + g_c(\boldsymbol{x}^{(t)}) \right) - \boldsymbol{x}^{(t)} \right\|^2$$

$$= \mathbb{E} \left\| \boldsymbol{x}_i^{(t,k-1)} - \boldsymbol{x}^{(t)} - \eta g_i(\boldsymbol{x}_i^{(t,k-1)}) - \eta g_c(\boldsymbol{x}^{(t)}) \right\|^2.$$

Separating mean and variance of the client gradient:

$$\mathbb{E} \left\| \boldsymbol{x}_i^{(t,k)} - \boldsymbol{x}^{(t)} \right\|^2 \leq \mathbb{E} \left\| \boldsymbol{x}_i^{(t,k-1)} - \boldsymbol{x}^{(t)} - \eta \nabla f_f(\boldsymbol{x}_i^{(t,k-1)}) - \eta g_c(\boldsymbol{x}^{(t)}) \right\|^2 + \eta^2 \sigma^2.$$

Next we use relaxed triangle inequality (Lemma D.2) to further separate terms:

$$\mathbb{E} \left\| \boldsymbol{x}_i^{(t,k)} - \boldsymbol{x}^{(t)} \right\|^2 \leq \left( 1 + \frac{1}{a} \right) \mathbb{E} \left\| \boldsymbol{x}_i^{(t,k-1)} - \boldsymbol{x}^{(t)} \right\|^2$$

$$+ (1+a)\eta^2 \mathbb{E} \left\| \nabla f_f(\boldsymbol{x}_i^{(t,k-1)}) + g_c(\boldsymbol{x}^{(t)}) \right\|^2 + \eta^2 \sigma^2$$

$$\leq \left( 1 + \frac{1}{a} \right) \mathbb{E} \left\| \boldsymbol{x}_i^{(t,k-1)} - \boldsymbol{x}^{(t)} \right\|^2$$

$$+ (1+a)\eta^2 \mathbb{E} \left\| \left( \nabla f_f(\boldsymbol{x}_i^{(t,k-1)}) - \nabla f_f(\boldsymbol{x}^{(t)}) \right) + \left( g_c(\boldsymbol{x}^{(t)}) - \nabla f_c(\boldsymbol{x}^{(t)}) \right) + \nabla f(\boldsymbol{x}^{(t)}) \right\|^2$$

$$+ \eta^2 \sigma^2$$

$$\leq \left( 1 + \frac{1}{a} \right) \mathbb{E} \left\| \boldsymbol{x}_i^{(t,k-1)} - \boldsymbol{x}^{(t)} \right\|^2$$

$$+ (1+a)2\eta^2 \underbrace{\mathbb{E} \left\| \nabla f_f(\boldsymbol{x}_i^{(t,k-1)}) - \nabla f_f(\boldsymbol{x}^{(t)}) \right\|^2}_{\mathcal{H}} + (1+a)4\eta^2 \mathbb{E} \left\| \nabla f(\boldsymbol{x}^{(t)}) \right\|^2$$

$$+ (1+a)4\eta^2 \underbrace{\mathbb{E} \left\| g_c(\boldsymbol{x}^{(t)}) - \nabla f_c(\boldsymbol{x}^{(t)}) \right\|^2}_{\mathcal{J}} + \eta^2 \sigma^2.$$

In the above inequality, term $\mathcal{H}$ can be converted via smoothness (Definition D.1), and term $\mathcal{J}$ is the variance of the centralized stochastic gradient (Equation 6). Letting $a = K$, we have:

$$\mathbb{E} \left\| \boldsymbol{x}_i^{(t,k)} - \boldsymbol{x}^{(t)} \right\|^2 \leq \left( \frac{K+1}{K} + 2K\eta^2\beta^2 \right) \mathbb{E} \left\| \boldsymbol{x}_i^{(t,k-1)} - \boldsymbol{x}^{(t)} \right\|^2 + 4K\eta^2 \mathbb{E} \left\| \nabla f(\boldsymbol{x}^{(t)}) \right\|^2$$

$$+ 4K\eta^2\sigma_c^2 + \eta^2\sigma^2.$$

Unrolling the above recurrence, we get:

$$\mathbb{E} \left\| \boldsymbol{x}_i^{(t,k)} - \boldsymbol{x}^{(t)} \right\|^2 \leq \left( 4K\eta^2 \mathbb{E} \left\| \nabla f(\boldsymbol{x}^{(t)}) \right\|^2 + 4K\eta^2\sigma_c^2 + \eta^2\sigma^2 \right) \sum_{j=0}^{k-1} \left( \frac{K+1}{K} + 2K\eta^2\beta^2 \right)^j$$

$$\leq \left( \frac{4\tilde{\eta}^2}{K\eta_s^2} \mathbb{E} \left\| \nabla f(\boldsymbol{x}^{(t)}) \right\|^2 + \frac{\tilde{\eta}^2}{K\eta_s^2} \left( \frac{1}{K}\sigma^2 + 4\sigma_c^2 \right) \right) \sum_{j=0}^{k-1} \left( \frac{K+1}{K} + \frac{2\tilde{\eta}^2\beta^2}{K\eta_s^2} \right)^j$$

Assuming $\eta_s \geq 1$, and $\tilde{\eta} \leq \frac{1}{18\beta}$, we have $\frac{K+1}{K} + \frac{2\tilde{\eta}^2\beta^2}{K\eta_s^2} \leq 1 + \frac{163}{162K}$, and hence

$$\sum_{j=0}^{k-1} \left(\frac{K+1}{K} + \frac{2\tilde{\eta}^2\beta^2}{K\eta_s^2}\right)^j \leq \sum_{j=0}^{K-1} \left(1 + \frac{163}{162K}\right)^j = \left(1 + \left(\frac{163}{162K}\right)^K - 1\right)\frac{162K}{163} \leq (e^{\frac{163}{162}} - 1)K \leq 2K.$$

$$\mathbb{E}\left\|\boldsymbol{x}_i^{(t,k)} - \boldsymbol{x}^{(t)}\right\|^2 \leq \left(\frac{2\tilde{\eta}}{9\beta K\eta_s^2}\mathbb{E}\left\|\nabla f(\boldsymbol{x}^{(t)})\right\|^2 + \frac{\tilde{\eta}^2}{K\eta_s^2}\left(\frac{1}{K}\sigma^2 + 4\sigma_c^2\right)\right)2K$$

Adding back the summation terms over $i$ and $k$, the bound on client drift is:

$$\mathcal{E}^{(t)} \leq \frac{4\tilde{\eta}}{9\beta\eta_s^2}\mathbb{E}\left\|\nabla f(\boldsymbol{x}^{(t)})\right\|^2 + \frac{2\tilde{\eta}^2}{\eta_s^2}\left(\frac{1}{K}\sigma^2 + 4\sigma_c^2\right).$$

$\square$

**Proofs of Theorem C.2 for Non-Convex Case**    Adding the statements of Lemmas D.12 and D.13, and assuming $\eta_s \geq \sqrt{S}$, we get:

$$\begin{aligned}
\mathbb{E}_{|t}\left[f(\boldsymbol{x}^{(t+1)})\right] &\leq f(\boldsymbol{x}^{(t)}) - \frac{4\tilde{\eta}}{9}\left\|\nabla f(\boldsymbol{x}^{(t)})\right\|^2 + \left(\frac{2}{KS}\sigma^2 + \sigma_c^2\right)\beta\tilde{\eta}^2 \\
&\quad + \frac{\beta}{27}\left(\frac{4\tilde{\eta}}{9\beta\eta_s^2}\mathbb{E}\left\|\nabla f(\boldsymbol{x}^{(t)})\right\|^2 + \frac{2\tilde{\eta}^2}{\eta_s^2}\left(\frac{1}{K}\sigma^2 + 4\sigma_c^2\right)\right) \\
&\leq f(\boldsymbol{x}^{(t)}) - \frac{1}{3}\tilde{\eta}\left\|\nabla f(\boldsymbol{x}^{(t)})\right\|^2 + \left(\frac{3}{KS}\sigma^2 + 2\sigma_c^2\right)\beta\tilde{\eta}^2
\end{aligned}$$

With the above, we have a recursive bound on the loss after round $t+1$. We can use lemmas (e.g., Lemma 2 in Karimireddy et al. (2020b), Lemma 4 in Stich (2019)) which establish a sub-linear convergence rate for such recursions. Assuming $\tilde{\eta} \leq \frac{1}{18\beta}$ and $\eta_s \geq \sqrt{S}$, we get:

$$\min_{t\in\{1,2,\ldots,T+1\}}\left\|\nabla f(\boldsymbol{x}^{(t)})\right\|^2 \leq \frac{54\beta F}{T+1} + \frac{6\sqrt{\left(\frac{3}{KS}\sigma^2 + 2\sigma_c^2\right)\beta F}}{\sqrt{T+1}}.$$

In the above expressions, $F$ is the error at initialization, $f(\boldsymbol{x}^{(0)}) - f(\boldsymbol{x}^*)$.

This yields an expression for the number of rounds $T$ to reach an error $\epsilon$:

$$T = \mathcal{O}\left(\frac{\left(\sigma^2 + KS\sigma_c^2\right)\beta F}{KS\epsilon^2} + \frac{\beta F}{\epsilon}\right).$$

