# OpenReview forum: "Mixed Federated Learning: Joint Decentralized and Centralized Learning"
_ICLR.cc/2023/Conference — Submitted to ICLR 2023_

### Official Review · Reviewer_oANg · 2022-10-16

**Confidence:** 4
**Correctness:** 2
**Technical Novelty And Significance:** 3
**Empirical Novelty And Significance:** 3
**Recommendation:** 3

**Clarity, Quality, Novelty And Reproducibility:**

### Clarity

- The paper is overall well-written (in terms of language) and easy to understand. The description of the 3 algorithms is particularly easy
 to follow given the color separation of the differences between these 3 algorithms and the baseline (federated averaging).

- The definitions of “centraloptimizer” and “mergeoptimizer” are missing. This is crucial for understanding the main functionality of the proposed algorithms. The authors provide an example for the mergeoptimizer (averaging) but do not discuss alternatives or how they would impact the performance.

- It is not clear how gradients are initialized for 2-way. There doesn't seem to be any related information in the supplementary material as well.

- The paper presents 3 methods for mixed FL. However, one of them (2-way) appears to be superior both in terms of convergence and empirical performance. It would be helpful if the authors highlight what is the merit of presenting and evaluating all 3 methods in the main paper (e.g., illustrate properties of the best performing method).

### Quality

- The paper is based in multiple parts on related work that is unavailable publicly (introduction, related work section, section 4.1, section 4.3). These references are redundant and authors should only cite publicly available work; otherwise any claims can be part of the paper under review.
  - The main motivation for the problem of performance degradation (communication + computation) is based on claims from such unavailable work ("Reducing Client Computation and Communication"). Therefore, there is a gap in understanding the importance of use-cases where mixed FL can improve the communication and computation performance.

- Figures 1,2 (smile) lack the FL baseline (i.e., train only on decentralized data).

- The evaluated approach for the bias mitigation (based on stack overflow + wikipedia) does not seem to reflect the motivating setup (high VS low end users).

- The predictive performance evaluation lacks baselines that also employ the centralized dataset/training (e.g., transfer learning).

### Novelty

- The idea of assisting federated learning given the presence of a centralized (non-private) dataset has been studied before in various works as the authors also discuss in their related work section.

- The proposed methods are novel but naturally build up on ways of combining gradients from different data distributions.

### Reproducibility

- The authors provide the code necessary to reproduce the experiments. The code is organized and documented.

- The authors provide details about the experimental setup (e.g., hyperparameters) in their supplementary material.


**Strength And Weaknesses:**

### Strengths

- The problem of distribution shift across federated learning clients is very important. The idea of tackling it by combining decentralized with centralized training is both practical and promising.

- The authors perform an empirical evaluation comparison based on 3 tasks with applications on federated learning.

- The authors discuss the convergence of the proposed solutions.

### Weaknesses

- The paper needs improvement in terms of clarity (see detailed review below).

- The convergence analysis assumes unbiased gradients (and thus iid distribution which is an invalid assumption on the federated learning setup). The authors also acknowledge this fact on their conclusion.

- The proposed methods require additional hyperparameters (ηc, ηm).

- The evaluation doesn't employ any baseline for the task of employing the centralized data (e.g., based on transfer learning).

- The use case of bias mitigation (section 4.2) is interesting; however there is a questionable (in terms of practicality) assumption on the existence of a non-private representative dataset for a subset of the clients.

- The paper does not assess the performance of the proposed methods in the presence of noise necessary to ensure differential privacy guarantees (given that federated learning on its own is not sufficient to provide strong privacy guarantees).




**Summary Of The Paper:**

The authors propose algorithms for federated learning (decentralized data) alongside the presence of a centralized dataset (termed "mixed federated learning"). The paper presents 3 algorithms based on gradient exchange between the clients and the server, benchmarks the corresponding performance on 3 tasks (simple classification, language modeling, movie recommendation) and theoretically analyzes the convergence properties. The benefits of the proposed methods are in terms of improved predictive power (due to the centralized data complementing the decentralized) and in terms of computation and communication efficiency (due to offloading intensive computations to the server side).

**Summary Of The Review:**

The paper is overall promising but requires a substantial amount of improvements mainly in terms of clarity and evaluation.
The novelty is limited but that is not an issue if the approach is clearly described and shown to be improving the state of the art. The paper would be substantially better in the presence of a convergence analysis that does not assume iid data (e.g., based on bounded dissimilarity assumptions).

The overall recommendation is a weak reject given the weaknesses of the work that are only partly addressable (given the necessary effort) with high confidence.

---

> ### Author Response · Authors · 2022-11-16
> **Rebuttal to Reviewer oANg**
>
> We thank the reviewer for noting that the paper is “overall well-written and easy to understand” and that “[the] description of the 3 algorithms is particularly easy to follow”.  We strove for clarity.
>
> In regards to a few specific deficiencies perceived by the reviewer:
>
> - While the convergence analysis does make some assumptions on the distribution of gradients, we note that these are not critical assumptions to the overall order of the convergence bound (we omit some in-depth math for brevity and space constraints).  Furthermore, we believe the agreement between experimental observations and theoretical convergence (as discussed in Section 5.3) clearly demonstrates that our convergence analysis is useful.
>
> - In regards to the comment on work that is ‘unavailable publicly’, these are merely instances where the citation would possibly reveal author identities, and so we anonymized in the spirit of ICLR’s double-blind review process.  The underlying papers are available publicly, and we will switch the references to standard (author-identity revealing) citations upon publication.  We feel that the motivations for mixed FL are compelling enough without needing details from these papers, but welcome further comments on how best to present these motivations.
>
> - Regarding comparing to transfer learning as a baseline, we have done exactly this in Appendix B.4.5 and Figure 13.
>
> - Our paper does not assess the performance of our algorithms under DP (e.g., DP-FedAvg) b/c our mixed FL algorithmic contributions are orthogonal to the application of DP (or any other variation of algorithm used for the federated portion of training).  The degradation of utility due to noising/clipping for DP will be no different than ‘standard’ FL.  We don’t present DP results as we feel this is unrelated and non-conflicting with the main contributions of our paper.  We realized that this aspect (the composability of mixed FL with any flavor of federated training) was understated in the previous draft, and we’ve now reworded a paragraph in the Conclusion to state this directly.
>
> - We’ve tweaked the Algorithms section for greater clarity (e.g, we now state that gradients are initialized to zeros in 2-way Gradient Transfer).
>
> - Regarding the evaluation approach for the language model/bias mitigation experiment (Section 4.2): we wanted to use datasets with are available open source, for reproducibility, hence the choice of federated Stack Overflow and Wikipedia.  These two datasets were also good choices because (as the experiments demonstrate) they differ significantly in word usage (a model trained purely on one performs poorly on the other, and vice versa). So we see no reason why they are not good stand-ins for differences in word usage b/w high and low-end mobile phone users.  We would love it if there was an available open source federated dataset of high and low-end phone usage!
>
> We thank the reviewer again for their time and engagement.

---

### Official Review · Reviewer_vzaj · 2022-10-24

**Confidence:** 3
**Correctness:** 3
**Technical Novelty And Significance:** 3
**Empirical Novelty And Significance:** 3
**Recommendation:** 6

**Clarity, Quality, Novelty And Reproducibility:**

Clarity & Quality: Quality of the results are adequate however, the clarity can be greatly improved by changing the presentation

Novelty: The proposed approach is novel and simple.

Reproducibility: The presentation and the details provided are reasonable to reproduce however having a bit more clarity will improve the reproducibility.


**Strength And Weaknesses:**

# Pros:

- The paper is well-written, easy to understand and follow along, however, I must admit that I did not pay an in depth attention to some of the theoretical analysis presented in section 5.

- Literature is covered adequately and the proposed paper is presented amongst the body of knowledge while it also contributes to extend the existing knowledge in the domain.

# Cons:

- Where are the references to the works in literature that showed the techniques such as using some portion of data at server to train the server model after aggregation, next another work that showed the linear combination of server and client models. Of course they fall under the category of “sequential central-then-FL” category as opposed to the joint FL in this paper.

- On page 2, “... large embedding table” is mentioned, does not make sense as to what it refers to at this point, it only makes sense after reading the contributions as one of the bullets talks about recommender systems.

- Also, the footnote 1 is actually the real difference from the so called sequential FL coined here vs the joint FL proposed, make this clear from the abstract itself rather than undermining it in the footnote.

- The notations in Algorithm 1 and 2 are never described but left for the reader to interpret.

- After looking at the results in Fig 1 and Fig 2, what is the point of having three variations especially all the three proposed algorithms more or less converge to the same accuracies in most of the cases.

- How movie recommendation embeddings are regularized is still unclear, please provide a formula on how this is done especially given the FL settings, even if the regularizations are there in literature. Maybe some reference to appendix also should make the concepts clear and how they play out in mixed FL.

- The way the paper presentation flows is a bit odd from the regular style of paper writing or formatting. For example, section 2 (algorithms) comes first and then section 3 (related work), then section 5 (convergence) comes after experiments, why not swap section 2 and 3 then place 5 as 4 and then experiments and so on.

- The empirical explanations in section 5.3 does not sound all that convincing, especially the trends of the Language modeling and movie recommendations are similar compared to the smile classification. The general common sense points to the fact that the celebA is custom partitioned and that probably is something to do with the observations in the paper. Did you rule that aspect out (or similar simple perspectives), before trying to prove the observations with such a complex empirical analysis?

- The FL settings such as the number of clients, the generation of non-IID distributions, optimizers user, etc are not explained.

- Also, the results show curves over a certain number of rounds that compare w.r.t the oracle. Obviously oracle has a single number, why not compare the end of training results with the oracle as well as the other state-of-the-art FL algorithms.


**Summary Of The Paper:**

The paper proposes a joint (mixed) FL strategy that jointly trains using a combination of centralized and federated models without transferring any data across server or client and vice versa.


**Summary Of The Review:**

Overall, the proposed approach, mixedFL sounds decent, however, the presentation can be improved further and also, the comparison with the state-of-the-art FL algorithms might help position the paper better as to how the proposed modifications benefit w.r.t the more adanced FL optimization algorithms.

---

> ### Author Response · Authors · 2022-11-16
> **Rebuttal to Reviewer vzaj**
>
> Thank you for your time and comments, they are very appreciated.  We’re pleased you found the paper “well-written, easy to understand and follow along”, that “the literature was covered adequately”, and you feel our work “contributes to extending the existing knowledge in the domain”.
>
> Our responses:
>
> We’re slightly confused on your comments that clarity should be improved (under ‘Clarity & Quality’) because in your earlier statements (under ‘Pros’) you state (as mentioned above) that the paper is well-written and easy to understand.  Can you help us better parse what you mean?
>
> You ask why we don’t compare with other ‘state-of-the-art’ FL algorithms (you don’t specify which).  A very important clarification: our mixed FL algorithms are *orthogonal to/composable with* any standard federated training approach. While we use the ubiquitous FedAvg in the paper, that’s just to focus on the things that are truly novel (the usage of an additional datacenter distribution/loss, the effects to convergence for different mixed FL algorithms, etc.).  We’ve now added a paragraph in the conclusion, describing how various other flavors of algorithm could be swapped into the federated portion of mixed FL.  E.g., a differentially private approach like DP-FedAvg could be used to increase privacy protection of federated data.
>
> You ask why we present 3 algorithms b/c they “… more or less converge to the same accuracies”.  We wish to clarify that they *do not always* converge the same (as observed both empirically and theoretically), and when and why they sometimes converge the same and sometimes don’t is a useful topic for ML practitioners adopting usage of our algorithms.  Furthermore, the algorithm which sometimes converges slowest (Parallel Training) also has the most desirable download bandwidth properties (Appendix A), so we wish to present all options to readers to inform them on tradeoffs b/w convergence and bandwidth.
>
> You mention some confusion with how our federated data was partitioned into clients.  E.g., you feel that the explanations given in Section 5.3 could be due to ‘the fact that the CelebA is custom partitioned’, and you ask why ‘the generation of non-IID distributions … [is] not explained”.  An important clarification: there is *no custom partitioning* of data taking place in any of these experiments; the federated dataset is ‘naturally occurring’. That is, data partitions by a naturally occurring client ID. In CelebA, it’s the celebrity identity (one client contains all photos of the same actress).  In Stack Overflow, it’s a particular S.O. user (one client contains all questions/answers authored by the same user).  So non-IIDness arises from the natural variation b/w humans (in appearance for CelebA, in writing style for S.0.).  This is why we chose (and like) these federated datasets, and why we have confidence that trends observed in simulations with these datasets accurately map to experiences in a real production (e.g., mobile phone) mixed FL setting.
>
> Section ordering:
> - We originally had Sections 4 and 5 in reverse order (as you propose), but felt the ‘higher order bit’ of relevance to audiences is that these algorithms work well in practice, as demonstrated by experiments.  With theoretical explanation for why different algorithms sometimes converge differently being a secondary interesting thing to cover.  We actually feel somewhat justified in this new ordering by something you mentioned: “the paper is well-written, easy to understand …, however, I must admit I did not pay in depth attention to … section 5.”  We feel the readability of the paper was aided by putting experiments before theory.
> - We put Algorithms (2) ahead of Related Work (3) because we felt that seeing our algorithms first gave useful context when making connections in Related Work. E.g., seeing 2-way Gradient Transfer first makes it easier to see the connections to e.g. SCAFFOLD for purely federated, non-IID clients. In our experience this is not an uncommon ordering?
>
> On notation and details:
> - You state “FL settings such as the number of clients, … are not explained”.  This was provided in Appendix B.2, for each experiment. You did make us realize we had neglected to mention optimizers used (SGD for all except for in Appendix B.4.4); we've corrected that now (thank you for pointing out).
> - Thanks for your comments on Algorithms 1/2, we have tweaked them a bit, to ensure that all notation is described (or easily inferable e.g. from context in a `for` loop).
>
> Finally, we appreciate your comments in regards to the key distinction being sequential mixing vs., joint mixing (footnote 1).  This is why we included Appendix B.4.5, to compare with sequential approach of transfer learning.  We really like your idea of putting the ‘sequential vs. joint’ distinction in the abstract; we're playing with a rewording to this effect right now.  We’ll respond with an update if we figure out a way without breaking page limits :-)
>
> Thank you!

---

### Official Review · Reviewer_YiaD · 2022-10-24

**Confidence:** 3
**Correctness:** 3
**Technical Novelty And Significance:** 2
**Empirical Novelty And Significance:** 2
**Recommendation:** 5

**Clarity, Quality, Novelty And Reproducibility:**

The paper is readable overall. However, the logic and presentation are not coherent in some places. For example, in Introdution section, the author mentions "Mixed loss f might be a more useful training objective for many reasons, including..." and then introduces two aspects including alleviating distribution skew and reducing communication and computation. From their statement, I cannot see that reducing communication and computation is directly related to this mixed loss. Although placing part of the computation of the embedding table on the server does reduce client-side communication and computation, the presentation and organization of this part may need to be reconsidered to highlight its relevance with the mixed loss. And the related work section is not very organized.


**Strength And Weaknesses:**

Strengths:
1. The general algorithm of Mixed FL is well constructed and seems valid.
2. The relevant experiments are relatively sufficient and able to reflect the effectiveness of proposed algorithm.

Weakness:
1.In 2-WAY GRADIENT TRANSFER, the client's gradient information will be passed to the server. However, issues related to data privacy during gradient tranmission do not appear to be explored in the paper.
2. Some technique behind the algorithm may not be that novel, such as computation offloading and gradient augmentation.


**Summary Of The Paper:**

To mitigate distribution shift and improve FL performance in inference scenarios, the paper proposes the Mixed FL, which adds centralized data to the server and co-trains a powerful and robust global model between clients and the server. And the 1/2-way gradient transfer could work to alleviate the influence caused by non-iid data. Thus, the model could perform better facing unfamiliar datasets with different distributions compared with clients' local datasets. Many experiments have been conducted to validate their algorithms in different domain tasks. Related theoretical explanations are provided for the convergence.


**Summary Of The Review:**

Overall, this work constructs a well-established algorithm for mixed FL. But some of its ideas are not very novel. This algorithm is more like integrating some existing technologies, such as compute offloading and gradient transferring. However, the experimental part is fully carried out, which can reflect the effectiveness of the algorithm. Moreover, the related theoretical proof is also relatively solid, although there are still some limitations. Again, further revisions are still required in the organization and presentation of some paragraphs.

---

> ### Author Response · Authors · 2022-11-16
> **Rebuttal to Reviewer YiaD**
>
> We thank the reviewer for analyzing our paper and providing useful feedback.
>
> Regarding the novelty of our work, we certainly agree that some constituent pieces have been used before in other contexts (which we clearly state in the Related Work, e.g., the relationship to work in addressing non-IID clients in standard FL).  But our contributions and algorithms are novel for the important problem context we are interested in (the example problems stated in Section 1 and experimentally demonstrated in Section 4).  No prior work has addressed our problem context, and our algorithms are demonstrated to perform effectively, so we feel that our work is useful to share with the FL community.
>
> In regards to a question on data privacy (e.g., during gradient transmission): *none* of our mixed FL algorithms require any additional data be transmitted from clients to server (above and beyond ‘standard’ FL). In particular in regards to 2-way Gradient Transfer, as described in Algorithm 1 and in further detail in Appendix A (Eqn 10), the information needed to construct the next round’s augmenting federated gradient ($\tilde{g}_f^{(t+1)}$) is already available in the clients’ model updates $\Delta_i^{(t)}$.  So there are no additional data privacy concerns on top of the standard concerns of FL.
>
> Furthermore, our mixed FL approach is *orthogonal* to the choice of algorithm used in the federated portion of mixed FL.  For this reason we didn’t spend much time on this subject (as our paper’s contributions lie in a different direction), but our mixed FL algorithms are naturally composable with differentially private manners of doing federated training.  We have added a paragraph in the conclusion which states this algorithmic orthogonality more explicitly.
>
> We thank the reviewer for the comments on clarity, and we reworded the ‘Reducing Client Computation and Communication’ paragraph to more clearly connect how the primary loss (at clients) and regularization (at datacenter) exactly maps to our mixed loss expression.

---

### Official Review · Reviewer_nSLv · 2022-10-25

**Confidence:** 2
**Clarity, Quality, Novelty And Reproducibility:** Clarity and quality require improveme…
**Correctness:** 2
**Technical Novelty And Significance:** 1
**Empirical Novelty And Significance:** 1
**Recommendation:** 3

**Strength And Weaknesses:**

In Section 2 Algorithms, it would be more helpful to explain the notation used and the various components in the two algorithms in this section rather than relegating details to the appendix. Additionally, it is not clear to me what the components ClientOptimizer, ServerOptimizer, CentralOptimizer, and MergerOptimizer in this context are, what each performs (they appear to be blackbox functions), how they relate to equation (1), and how they connect to Section 5.

The rationale on the StackOverflow and Wikipedia (and the allocation of 25% server, 75% client) dataset as it pertains to client devices is not thoroughly explained in the experimental section.

Section 4.3 as it pertains to "mixing different loss" and to the focus of the paper is not clear to me.

**Summary Of The Paper:**

The paper focuses on a federated learning setting where there is a form of potentially high bias on the client-side (I use the work bias in the basic sense (i.e., favored towards to) instead of the pure statistical sense). These biases are exemplified by the datasets used by the paper, which come in the form of unbalanced labeling on client devices, and differences in the specification of client devices (i.e., high-end phones vs low-end phones). For these biases, the paper proposes to mix the datasets between the server (or datacenter) and the clients. Specifically, the paper uses two datasets:

* CelebA: the dataset is partitioned into people who are "smiling" and who are "not smiling." The claim is that the majority of pictures on people's phone are smiling. Here, the server contains the non-smiling pictures and the clients contain smiling pictures and the task is to predict smiling or not smiling.

* StackOverflow and Wikipedia: the StackOverflow dataset is used for clients, and the Wikipedia dataset is used for the server.

The paper also proposes a mixed loss function, which is the sum of the loss on server data and the loss on client data.



**Summary Of The Review:**

The proposed idea and presentation require more work.

Update: Thank you to the authors for their reply. I will maintain my review.

---

> ### Author Response · Authors · 2022-11-16
> **Rebuttal to Reviewer nSLv**
>
> Thank you for your review, and very useful individual comments (which led to several changes in wording). We address them in turn, below, after first addressing your high-level feedback.
>
> We disagree (politely but very strongly) with your view of ‘Correctness’, that our paper’s claims are incorrect or not well-supported.  There is no substantiation of this in your other comments.  Can you please clarify which claims you feel are in error?  We feel we’ve provided both empirical and theoretical support for our claims in the paper, but would be happy to address specific concerns of yours.
>
> We respectfully but similarly disagree with your view of novelty/significance: we’re not aware of any prior work that has mixed federated and centralized learning to this end (and we rather exhaustively covered things in ‘Related Work’, as noted by other Reviewers).
>
> Specific comments:
>
> Thank you for your thoughts on the Algorithms section.  We should have been more clear in connecting the gradient (`g`) calculations of Algos 1/2 with their respective loss (`f`) terms of Equations 2 and 3.  We’ve tweaked Algos 1/2 to better convey the relationship.  While we are decidedly space-constrained, we did strive to have the Algorithms section stand on its own (e.g., Algorithms 1 and 2 introduce all hyperparameters, etc.).  The only aspects that are relegated to Appendix B.2 are the specific hyperparameter selections made for the various experiments in Section 4.
>
> The optimizers are indeed treated as black box functions, as their selection is largely orthogonal to the contributions we make in this paper. We added clarification in Appendix B.2 that these are SGD for all experiments (except for Appendix B.4.4 where we demonstrate use of 1-way Grad Transfer with FedADAM).
>
> Thank you for your comments in regards to Section 4.3; we’ve reworded it (as well as some rewordings in Table 1 and Section 2) to communicate more clearly the following:
> - Mixed FL involves 2(+) losses, which differ significantly
> - This difference can be in distributions that respective batches are drawn from (e.g., smiles vs non-smiles, Wikipedia words vs. StackOverflow words, etc.), *and/or* …
> - This difference can be in the functional form of the loss terms (e.g., a hinge function for federated client losses, a spreadout regularization term for centralized loss).
>
> The point of Section 4.3 is to demonstrate an example of this latter, distinct use case, and how mixed FL successfully encompasses this use case as well.
>
> Thank you again for a very helpful review, your thoughts helped us to more clearly convey important details.

---

### Author Response · Authors · 2022-11-15
**Note in delay in rebuttals**

Reviewers,

We apologize for the delay in sending out our initial rebuttals; some personal and professional complications impeded the ability of lead author to respond last week.  We will be sending our rebuttals today.  We thank the reviewers for the time spent in writing reviews, and wanted to make sure they knew their reviews were not written in vain.

Regards,
The authors

---

### Author Response · Authors · 2022-11-19
**Rebuttal summary**

We regret that none of our reviewers responded to our rebuttals, as we feel there were a number of misperceptions which we addressed.  E.g, the novelty of our work (joint federated/centralized training has never been described before), or the fact that our algorithms are general and easily compose with various kinds of federated learning algorithms (e.g., using FedAdam or DP-FedAvg instead of FedAvg). We also pointed our reviewers to parts they had missed (e.g., the comparison with transfer learning in B.4.5; the experiment configurations in B.2).

We do greatly appreciate the reviewers' first round of reviews, as they pointed out several ways to make the exposition clearer (e.g., missing variable introductions in Algorithms 1/2, etc.).  We made several edits to the paper to better convey things.

---

### Decision · Program_Chairs · 2023-01-20

**Decision:**

Reject

**Justification For Why Not Higher Score:**

(1) limited novelty
(2) more clarity on the writing is needed
(3) the convergence analysis used overly simplified iid assumption which is not standard for FL

**Justification For Why Not Lower Score:**

N/A

**Metareview: Summary, Strengths And Weaknesses:**

This paper studies an interesting setting for federated learning where there exists a shift of distribution from the client side to the server side. This setting is practically important and the proposed solution of a joint mixed strategy that joints trains  a more powerful global model between clients and server by adding centralized data to the server.   The authors provide  organized & documented code and the experiments appear reproducible. The main criticisms from reviewers include (1) limited novelty (2) more clarity on the writing is needed (3) the convergence analysis used overly simplified iid assumption which is not standard for FL, as typical FL analysis uses the noniid bounded difference assumption. Overall, this paper is promising and the authors are encouraged to invest some non-trivial efforts to significantly improve the paper.